# FedCF: Fair Federated Conformal Prediction

## Abstract

Conformal Prediction (CP) is a widely used technique for quantifying uncertainty in machine learning models. In its standard form, CP offers probabilistic guarantees on the coverage of the true label, but it is agnostic to sensitive attributes in the dataset. Several recent works have sought to incorporate fairness into CP by ensuring conditional coverage guarantees across different subgroups. One such method is Conformal Fairness (CF). In this work, we extend the CF framework to the Federated Learning setting and discuss how we can audit a federated model for fairness by analyzing the fairness-related gaps for different demographic groups. We empirically validate our framework by conducting experiments on several datasets spanning multiple domains, fully leveraging the exchangeability assumption.

## 1 Introduction

Ensuring model fairness is a critical thrust of trustworthy machine learning (ML). ML models, when not calibrated for fairness, are prone to developing biases at each stage of an ML pipeline, as reflected by their predictions Mehrabi et al. (2021). We define bias as disparate performance (i.e., accuracy for classification) between different sub-populations. In the data collection phase, measurement bias may occur due to disproportionate data collection on sub-populations, while representation bias manifests from a lack of training data on specific strata. During training, these biases are inductively learned by the model–leading to incorrect predictions in safety-critical tasks. These models are also susceptible to algorithmic bias, resulting from regularization and optimization techniques during model training, which incorrectly generalize for marginalized groups. To mitigate these risks, many ML models must adhere to regulations placed by local governing bodies (Hirsch et al., 2023). Towards model compliance, Komala et al. (2024); Agrawal et al. (2024); Jones et al. (2025) have proposed approaches to enhance model fairness in varying tasks, including federated graph learning and representation learning.

Developing robust ML frameworks with mathematically rigorous guarantees is also essential for building actionable, trustworthy ML models for safety-critical tasks. In this frontier, researchers have increasingly explored Conformal Prediction (CP)–an uncertainty quantification (UQ) technique that only assumes statistical exchangeability–to develop trustworthy ML models (Vovk et al., 2005). Unlike traditional point-wise prediction in ML, CP guarantees that the correct outcome will be in a prediction set with a user-specified property. Practitioners have adopted CP due to its model-free assumption and post-hoc application (Cherian & Bronner, 2020). Additionally, users can apply CP to structured data (such as graphs), which cannot be used with traditional IID-based methods (Maneriker et al., 2025). However, vanilla CP is not calibrated for fairness and can be inherently unfair Cresswell et al. (2025).

Several approaches have been proposed at the intersection of fairness and CP–each catering to different tasks and notions of fairness. Romano et al. (2020a) developed a CP approach for the regression setting to ensure equalized coverage across protected groups. Lu et al. (2022) considers equalized coverage in a classification task for medical imaging. Zhou & Sesia (2024) extends Romano et al. (2020a) and provides an algorithm adaptive to sensitive groups to increase the predictive power of the CP sets when several sensitive attributes are present, and focuses on the classification task. Lastly, Vadlamani et al. (2025) provides a framework to ensure fair coverage of positive outcomes, without requiring protected attributes at inference time, unlike prior work. Orthogonally, CP has been used to enhance the fairness of other tasks. To mitigate bias in LLM-based recommender systems, Fayyazi et al. (2025) explores iteratively using fairness-aware CP.

While there are several approaches to integrating fairness into CP, these methods are not considered when the training data is decentralized (i.e., available only to clients) and the ML model is stored on a centralized server. Extending these CP methods to the federated learning (FL) setting is essential because tasks that benefit from fair uncertainty quantification (such as those in healthcare and finance) also often have privacy considerations, making it infeasible to keep data on a centralized server. Thus, we extend the work of Vadlamani et al. (2025) using recent literature in Federated CP (Lu et al., 2023).

**Key Contributions**: We develop Federated Conformal Fairness (FedCF) and extend the Conformal Fairness (CF) framework (Vadlamani et al., 2025) to the FL setting while maintaining the theoretical fairness guarantees provided by CF.

**Extending CF Theory to FL.** We discuss how to bound conditional coverage according to a user-specified fairness notion when data is decentralized. To facilitate this, we provide a sufficient set of terms that a client can compute using local data, how the server should aggregate these terms to bound the conditional coverage, and theoretically prove the validity of our approach.

**Descent-Based CF Formulation.** We revise the original CF algorithm to reduce the number of communication rounds required to construct a fair conformal predictor. We do this by reformulating the original iterative approach presented in Vadlamani et al. (2025) into a descent-based optimization framework. This allows FedCF to be embedded directly into an FL-style algorithm and to integrate naturally with existing FL systems.

**Flexible Aggregation Protocols.** We consider the client-server communication overhead and its tradeoff with preserving data privacy. Specifically, we propose two approaches, with one having less communication overhead and the other being more privacy-preserving of client data.

**Real-World Empirical Validation.** We evaluate FedCF on several datasets with naturally induced *data heterogeneity*, including tabular, graph, and image datasets, and for multiple popular fairness metrics, and observe that FedCF can control for a particular coverage gap level while maintaining the original CP coverage guarantee.

## 2 BACKGROUND

### 2.1 CONFORMAL PREDICTION

Conformal Prediction (CP) (Vovk et al., 2005) is a widely used framework for quantifying predictive uncertainty in ML. CP provides rigorous statistical guarantees without imposing assumptions on the model, requiring only that the data are *exchangeable*–a broader condition than IID and compatible with non-IID or structured settings (e.g., graphs).

We focus on split (inductive) CP in the classification setting. Let $\boldsymbol{x}_i \in \mathcal{X} = \mathbb{R}^d$ and $y_i \in \mathcal{Y} = \{0, \ldots, C-1\}$ denote features and labels. Given a calibration dataset, $\mathcal{D}_{\text{calib}} = \{(\boldsymbol{x}_i, y_i)\}_{i=1}^n$, our goal is to construct a set-valued predictor $\mathcal{C}$ such that, for an exchangeable test point $(\boldsymbol{x}_{\text{test}}, y_{\text{test}})$,

$$1 - \alpha \leq \Pr[y_{\text{test}} \in \mathcal{C}(\boldsymbol{x}_{\text{test}})] \leq 1 - \alpha + \frac{1}{|\mathcal{D}_{\text{calib}}|}, \tag{1}$$

where $1 - \alpha \in (0, 1)$ is the target *coverage level*. We refer to Equation 1 as the *coverage guarantee*. Concretely, given a non-conformity score $s : \mathcal{X} \times \mathcal{Y} \to \mathbb{R}$, define the *conformal quantile* as

$$\hat{q}(\alpha) = \text{Quantile}\left( \frac{\lceil (n+1)(1-\alpha) \rceil}{n}; \{s(\boldsymbol{x}_i, y_i)\}_{i=1}^n \right).$$

The resulting prediction set $\mathcal{C}_{\hat{q}(\alpha)}(\boldsymbol{x}_{\text{test}}) = \{y \in \mathcal{Y} : s(\boldsymbol{x}_{\text{test}}, y) \leq \hat{q}(\alpha)\}$ satisfies the guarantee in 1.

**Evaluating CP**: Two standard metrics are used: (1) *Coverage*, the estimated test-time probability, $\Pr[y_{\text{test}} \in \mathcal{C}_{\hat{q}(\alpha)}(\boldsymbol{x}_{\text{test}})]$; and (2) *Efficiency*, the average prediction set size, $|\mathcal{C}_{\hat{q}(\alpha)}(\boldsymbol{x}_{\text{test}})|$. These are typically in tension as achieving a higher desired coverage often necessitates larger sets.

### 2.2 FEDERATED LEARNING

A key contributor to developing strong deep learning models is providing a large amount of quality training data (Kaplan et al., 2020). However, in certain domains, such as healthcare and finance, collecting large amounts of data may be prohibitive due to privacy concerns. Federated

learning (FL) McMahan et al. (2017) is a framework for collaborative learning that keeps training data decentralized and private. Given $K$ clients that will participate in training, each client has its own training data that it wants to keep private. The goal is to optimize a global loss function, $L$, that is the weighted average of local risk functions, $\ell_k$. Formally, FL finds weights $\theta^*$ s.t. $\theta^* = \arg\min_\theta \left\{ L(\theta) = \sum_{k=1}^{K} w_k \cdot \mathbb{E}_{(x^{(k)}, y^{(k)}) \sim P_k} \left[ \ell_k \left( \theta; x^{(k)}, y^{(k)} \right) \right] \right\}$, where $P_k$ is client $k$'s local distribution and $\boldsymbol{w} \in \Delta^k$ are weights (Lu et al., 2023).

## 2.3 Federated Conformal Prediction (FCP)

**Setting**: In FL, the development and calibration datasets are partitioned over $K$ clients. Meaning, each client $k \in \{1, \dots, K\} = \mathcal{K}$ retains a private calibration set $\mathcal{D}_{\text{calib}}^{(k)} = \left\{ \left( \boldsymbol{x}_i^{(k)}, y_i^{(k)} \right) \right\}_{i=1}^{n_k}$ drawn from an unknown local distribution $P_k$. The goal is to still construct a prediction set function $\mathcal{C}$ such that for any test point $(\boldsymbol{x}_{\text{test}}, y_{\text{test}}) \sim Q_{\text{test}}$, where $Q_{\text{test}} = \sum_{k=1}^{K} \gamma_k P_k$ is the mixutre distribution with weights $\gamma_k \propto (n_k + 1)$ (Lu et al., 2023), Equation 1 is satisfied. This is done while respecting the communication and privacy constraints of FL.

**Partial exchangeability and the FCP algorithm.** Lu et al. (2023) introduce partial exchangeability: within each client, the multiset $\left\{ s\left( \boldsymbol{x}_1^{(k)}, y_1^{(k)} \right), \dots, s\left( \boldsymbol{x}_{n_k}^{(k)}, y_{n_k}^{(k)} \right), s(\boldsymbol{x}_{\text{test}}, y_{\text{test}}) \right\}$ is exchangeable with probability $\gamma_k$. Under this assumption, the FCP method aggregates all non-conformity scores, orders them, and selects the $(1 - \alpha)(N + K)$-th statistic as follows

$$\hat{q}(\alpha) = \text{Quantile}\left( \frac{\lceil (N + K)(1 - \alpha) \rceil}{N}; \left\{ \left( \boldsymbol{x}_i^{(k)}, y_i^{(k)} \right) \right\}_{k,i} \right)$$

where $N = \sum_{k=1}^{K} n_k$. The prediction set $C_\alpha(\boldsymbol{x}) = \{ y : S(\boldsymbol{x}, y) \leq \hat{q}_\alpha \}$ then satisfies

$$1 - \alpha \ \leq \ \Pr[y_{\text{test}} \in C_\alpha(\boldsymbol{x}_{\text{test}})] \ \leq \ 1 - \alpha + \frac{K}{N+K}. \tag{2}$$

**Communication-efficient quantile sketches.** To preserve privacy, instead of transmitting all $N$ scores to the server, each client can send a mergeable sketch (e.g., using T-Digest (Dunning, 2021) or DDSketch (Masson et al., 2019)). Doing so will loosen the guarantee given in Equation 2.

## 2.4 Conformal Fairness

While CP provides marginal coverage guarantees, it is agnostic to sensitive attributes within the data. So different groups can receive systematically different coverages. The Conformal Fairness (CF) framework (Vadlamani et al., 2025) formalizes the notion of fairness for prediction sets by considering the disparity in conditional coverage between sensitive groups, all while retaining the validity based on the CP exchangeability. At a high level, CF adapts group-fairness notions (e.g., Demographic Parity and Equal Opportunity) to the set-valued outputs of CP and then tunes a score threshold to satisfy a user-specified *"closeness" criterion*, $c$, on inter-group disparities. Using the exchangeability assumption, CF can be applied to non-IID/structured data.

**From point predictions to set-based fairness.** Let $\mathcal{C}_\lambda(\boldsymbol{x}) = \{ y \in \mathcal{Y} : s(\boldsymbol{x}, y) \leq \lambda \}$ denote the CP prediction set at score threshold $\lambda$. CF adapts classical group-fairness metrics by replacing point-prediction events (i.e., $\tilde{y} = \hat{Y}$) with set-membership events (i.e., $\tilde{y} \in \mathcal{C}_\lambda(X)$) and evaluating disparities across groups $\mathcal{G}$ and, when appropriate, advantaged labels $\mathcal{Y}^+$. For example, a set-based Demographic Parity-style constraint can be written as,

$$\left| \Pr[\tilde{y} \in \mathcal{C}_\lambda(X) \mid X \in g_a] - \Pr[\tilde{y} \in \mathcal{C}_\lambda(X) \mid X \in g_b] \right| \ \leq \ c \quad \forall g_a, g_b \in \mathcal{G}, \ \tilde{y} \in \mathcal{Y}^+,$$

with analogous set-based forms for other common group-fairness metrics.

**Conditional coverage as the fairness control knob.** To evaluate a chosen fairness notion, CF filters the calibration data to the relevant subpopulation (e.g., a group or a group-and-label slice) via a filter function, $F_m$, and uses conditional coverage estimates under that filter. It then searches a threshold space $\Lambda$ to identify $\lambda_{\text{opt}}$ that satisfies the closeness criterion across groups (and labels, if required) while maintaining CP validity. A key technical ingredient is that CP coverage holds when labels are fixed to a particular $\tilde{y}$, which underpins group- and class-conditional control in CF.

**Guarantees and trade-offs.** CF provides a theoretically grounded procedure to bound fairness disparities (as defined above) without sacrificing CP's finite-sample coverage guarantees, which is

Table 1: Important notation used for coverage gap calculation.

| Notation | Definition | | Notation | Definition |
|---|---|---|---|---|
| $n_k$ | $\left\| \mathcal{D}_{\text{calib}}^{(k)} \right\|$ | | $\mathcal{D}_{\text{calib}}^{(k)}$ | Client $k$'s calibration dataset. |
| $n_k^{(g,\tilde{y})}$ | $\left\| \mathcal{S}_k^{(g,\tilde{y})} \right\|$ | | $\mathcal{S}_k^{(g,\tilde{y})}$ | $\left\{ (\boldsymbol{x}_i, y_i) \in \mathcal{D}_{\text{calib}}^{(k)} \mid F_M(\boldsymbol{x}_i, y_i, g, \tilde{y}) = 1 \right\}$ |
| $\gamma_k$ | $\Pr[E_k]$ | | $E_k$ | The event $\boldsymbol{x}_{\text{test}}$ is exchangeable with $\mathcal{D}_{\text{calib}}^{(k)}$. |
| $\pi^{(g,\tilde{y})}$ | Point estimate for Term $\text{(IV)}$ in Equation 3. | | $L^{(g,\tilde{y})}, U^{(g,\tilde{y})}$ | Bounds for Term $\text{(IV)}$ in Equation 3. |
| | | | $\alpha_k^{(g,\tilde{y});\lambda}$ | $\sum\limits_{(\boldsymbol{x}_i,\text{-}) \in \mathcal{S}_k^{(g,\tilde{y})}} \mathbf{1}[s(\boldsymbol{x}_i, \tilde{y}) \leq \lambda]$ |

empirically backed by CF reducing fairness violations across several metrics and remains effective with multiple sensitive attributes (intersectional groups). In practice, satisfying stricter fairness closeness $c$ increases the average prediction set size, reflecting the fairness–efficiency trade-off.

**Practical advantages.** Unlike many conditional-CP baselines that require group membership at inference time or are model-specific, CF's set-based metrics and thresholding procedure do not require protected attributes at test time and apply across different non-conformity scores and data modalities, making it compatible with downstream deployment constraints.

## 3 FedCF Theory and Methodology

In this section, we begin by establishing the theoretical and methodological foundations. We first redefine the concept of the *coverage gap* (3.1) and introduce a descent-based reformulation of the CF Framework (3.2), both within the federated setting. Following this, we present the Federated Conformal Fairness (FedCF) Framework.

### 3.1 FedCF: Extending Coverage Gap to the Federated Setting

Let $F_M$ be a filter function for some fairness metric. Then, we define the fairness-specific coverage level for positive label $\tilde{y} \in \mathcal{Y}^+$ in group $g \in \mathcal{G}$ as $\Pr[s(\boldsymbol{x}_{\text{test}}, \tilde{y}) \leq \lambda \mid F_M(\boldsymbol{x}_{\text{test}}, y_{\text{test}}, g, \tilde{y}) = 1]$. In the federated setting, since the data is decentralized, we cannot directly estimate this quantity. To address this, we rewrite the quantity–using notation in Table 1–as,

$$\Pr[s(\boldsymbol{x}_{\text{test}}, \tilde{y}) \leq \lambda \mid F_M(\boldsymbol{x}_{\text{test}}, y_{\text{test}}, g, \tilde{y}) = 1] = \sum_{k=1}^{K} \Big( \underbrace{\Pr[s(\boldsymbol{x}_{\text{test}}, \tilde{y}) \leq \lambda \mid F_M(\boldsymbol{x}_{\text{test}}, y_{\text{test}}, g, \tilde{y}) = 1, E_k]}_{\text{①}}$$

$$\cdot \underbrace{\Pr[F_M(\boldsymbol{x}_{\text{test}}, y_{\text{test}}, g, \tilde{y}) = 1 \mid E_k]}_{\text{②}} \cdot \underbrace{\Pr[E_k]}_{\text{③}} \Big) \cdot \Big( \underbrace{\Pr[F_M(\boldsymbol{x}_{\text{test}}, y_{\text{test}}, g, \tilde{y}) = 1]}_{\text{④}} \Big)^{-1}. \quad (3)$$

With this reformulation, we can estimate different terms individually—either locally on each client or globally on the server. The following theorem presents two types of fairness-specific conditional coverage estimates: (1) **interval bounds** and (2) **point estimates**. The estimates for the individual terms are provided in Lemmas B.1, B.2, and B.3 in Appendix B. For clarity, Table 1 summarizes the primary notation used in our main theorem. A full notations table can be found in Appendix A.

**Theorem 3.1.** *The fairness-specific coverage level (Equation 3) can be bounded as*

$$L_{cov}(\lambda, F_M, g, \tilde{y}) \leq \Pr[s(\boldsymbol{x}_{test}, \tilde{y}) \leq \lambda \mid F_M(\boldsymbol{x}_{test}, y_{test}, g, \tilde{y}) = 1] \leq U_{cov}(\lambda, F_M, g, \tilde{y}),$$

*where*

$$L_{cov}(\lambda, F_M, g, \tilde{y}) = \sum_{k=1}^{K} \frac{\gamma_k \alpha_k^{(g,\tilde{y});\lambda} n_k^{(g,\tilde{y})}}{(n_k^{(g,\tilde{y})}+1)(n_k+1)U^{(g,\tilde{y})}} \text{ and } U_{cov}(\lambda, F_M, g, \tilde{y}) = \sum_{k=1}^{K} \frac{\gamma_k(\alpha_k^{(g,\tilde{y});\lambda}+1)}{(n_k+1)L^{(g,\tilde{y})}}. \quad (4)$$

*If the data is IID, using MLE estimates for each term, we get the following estimate for the fairness-specific coverage level*

$$\Pi_{cov} = \Pr[s(\boldsymbol{x}_{test}, \tilde{y}) \leq \lambda \mid F_M(\boldsymbol{x}_{test}, y_{test}, g, \tilde{y}) = 1] = \sum_{k=1}^{K} \frac{\gamma_k \alpha_k^{(g,\tilde{y});\lambda}}{n_k \pi^{(g,\tilde{y})}}. \tag{5}$$

Theorem 3.1 gives us bounds for the coverage level, which we can convert into bounds for the coverage gap between groups for fairness evaluation. The interval bounds provide finite sample guarantees that are typically seen in the CP literature, at the cost of being a more conservative estimate. Conversely, point estimates provide tight coverage estimates, but assume IID data, and may violate those guarantees.

## 3.2 FedCF: Revisiting the Conformal Fairness Algorithm

---

**Algorithm 1** Descent-Based CF Optimization

---

1: **procedure** Fair_Opt_Descent($\tilde{y}$, $\lambda_0$, $\mathcal{G}$, $c$, $F_M$, num_rounds, $\eta$, $\mu$)
2:     $\lambda_{\text{opt}} = 1$
3:     **for** ($t = 0$; $t{+}{+}$; $t <$ num_rounds) **do**
4:         $cg_t =$ coverage_gap($\lambda_0, F_m, \tilde{y}, \mathcal{G}$)
5:         **if** $cg_t \leq c$ **and** $t = 0$ **then**
6:             **return** $\lambda_0$
7:         **else if** $cg_t \leq c$ **and** $\lambda_t < \lambda_{\text{opt}}$ **then**
8:             $\lambda_{\text{opt}} = \lambda_t$
9:         **end if**
10:       $b_{t+1} = \mu \cdot b_t + (cg_t - c)$
11:       $\Delta\lambda = (\lambda_{opt} - \lambda_t)\mathbf{1}_{[b_{t+1} \geq 0]}$
              $+ (\lambda_t - \lambda_0)\mathbf{1}_{[b_{t+1} < 0]}$
12:       $p_t = \max\{\lceil \log_2(\frac{\eta}{\Delta\lambda}) \rceil, 0\}$
13:       $\eta_t =$ update_lr($\eta, p_t, b_{t+1}$)
14:       $\lambda_{t+1} = \lambda_t + \eta_t b_{t+1}$
15:     **end for**
16:     **return** $\lambda_{\text{opt}}$
17: **end procedure**

---

One drawback of the original Conformal Fairness algorithm is that it creates a sampled, discretized search space and iterates to find the minimal $\lambda$ satisfying the fairness specification. This process requires computing the coverage gap every iteration and for each positive label, which becomes even more inefficient in the federated setting, where coverage gap computation requires client-server communication.

Algorithm 1 describes the core descent-based CF algorithm. [1] The algorithm takes as input the calibration set for each client, $\{\mathcal{D}_{\text{calib}}^{(k)}\}_{k \in \mathcal{K}}$, the set of (positive) labels, the set of sensitive groups, $\mathcal{G}$, a closeness criterion, $c$, and a filtering function, $F_M$. Additionally, we initialize a threshold, $\lambda_0$, as the $\hat{q}$ value given by the FCP algorithm (Lu et al., 2023), to ensure $1 - \alpha$ coverage is still satisfied.

For a given threshold $\lambda_t$, we can compute the coverage gap $cg_t$ and evaluate whether it adheres to our fairness constraint. FedCF solves, $\lambda_{opt} = \min_{\lambda \in \Lambda} \lambda$, subject to $cg_t - c \leq 0$. We solve this using a framework analogous to Gradient Descent (GD) with Momentum (Polyak, 1964). Let $\eta$ and $\mu$ be the initial learning rate and momentum constant, respectively. The update rule for $\lambda_t$ is,

$$\lambda_{t+1} = \lambda_t + \eta_t \cdot b_{t+1} = \lambda_t + \eta \cdot (1/2)^{p_t} \cdot b_{t+1} \in [\lambda_0, \lambda_{opt}],$$

where $b_{t+1} = \mu \cdot b_t + (cg_t - c)$ is the modified step size, $p_t = \max\{\lceil \log_2(\frac{\eta}{\Delta\lambda}) \rceil, 0\}$ determines the scaling factor to update $\eta$ by and ensure $\lambda_{t+1} \in [\lambda_0, \lambda_{opt}]$ where $\Delta\lambda = (\lambda_{opt} - \lambda_t)\mathbf{1}_{[b_{t+1} \geq 0]} + (\lambda_t - \lambda_0)\mathbf{1}_{[b_{t+1} < 0]}$.

We do not stop immediately once a satisfactory $\lambda$ is found; instead, we continue exploring to check whether a smaller $\lambda$ exists. This algorithm directly applies to the federated setting, with one important consideration: the computation of the coverage gap in Line 4.

## 3.3 FedCF: the end-to-end Federated Conformal Fairness framework

Having established the sufficient terms to compute the fairness-specific coverage gap, we now present the FedCF framework. We discuss FedCF in the context of the interval-bounds estimates from Theorem 3.1, noting the discussion also applies to the point-estimate case by setting

---

[1]We omit the iteration over the positive labels for brevity and present just the core optimization.

$L_{\text{cov}} = U_{\text{cov}} = \Pi_{\text{cov}}$. The fairness-specific coverage gap is given by,

$$\text{cg}(\lambda, F_M, \tilde{y}, \mathcal{G}) := \max_{g_a \in \mathcal{G}}\{U_{\text{cov}}(\lambda, F_M, g_a, \tilde{y})\} - \min_{g_b \in \mathcal{G}}\{L_{\text{cov}}(\lambda, F_M, g_b, \tilde{y})\} \quad (6)$$

$$= \max_{g_a, g_b \in \mathcal{G}}\{U_{\text{cov}}(\lambda, F_M, g_a, \tilde{y}) - L_{\text{cov}}(\lambda, F_M, g_b, \tilde{y})\}. \quad (7)$$

While equations 6 and 7 are mathematically equivalent, their formulations lead to two different communication and aggregation strategies demonstrating the tradeoff between **communication overhead** and **privacy**. We present the *communication efficient* protocol in the main paper and the *enhanced privacy* protocol in Appendix D. In Appendix D, we also present a hybrid protocol, where clients select whether to use the *communication efficient* or *enhanced privacy* protocol. We include FedCF extensions concerning differential privacy in Appendix F.

Note that $U_{\text{cov}}$ and $L_{\text{cov}}$ depend on $L^{(g,\tilde{y})}$ and $U^{(g,\tilde{y})}$, respectively. Since these quantities are also computed in a federated manner on the server, we compute them *prior* to computing the coverage gap for any particular $\lambda^2$. Given that these priors are available on the server, we can compute the fairness-specific coverage gap. From Theorem 3.1, each client computes and sends two values for each $(g, \tilde{y}) \in \mathcal{G} \times \mathcal{Y}^+$ pair: $\frac{\alpha_k^{(g,\tilde{y});\lambda} \cdot n_k^{(g,\tilde{y})}}{\left(n_k^{(g,\tilde{y})}+1\right) \cdot (n_k+1)}$ and $\frac{\alpha_k^{(g,\tilde{y});\lambda}+1}{n_k+1}$. Once the server receives these pairs from each client, it proceeds to aggregate these quantities to derive $L_{\text{cov}}$ and $U_{\text{cov}}$ for each $(g, \tilde{y})$. $U_{\text{cov}}$ is limited 1 to reconcile $\Pr[\cdot] \leq 1$ for any event. The final coverage gap is determined as per Equation 6. Algorithms 2 and 3 describe the federated coverage gap algorithm.

**Communication Complexity and Privacy Implications.** Each client is responsible for sending messages of size totaling $\mathcal{O}(2 \cdot |\mathcal{G}||\mathcal{Y}^+|)$ to the server per server round. While this is linear in terms of the number of $(g, \tilde{y})$ pairs, we note that with enough $\lambda$s, the server can learn the distribution of $\Pr[s(\boldsymbol{x}_{\text{test}}, \tilde{y}) \leq \lambda \mid F_M(\boldsymbol{x}_{\text{test}}, y_{\text{test}}, g, \tilde{y}) = 1, E_k]$.

---

**Algorithm 2** Server-side Aggregation for Coverage Gap

1: **procedure** SERVERCG($\lambda, F_M, \tilde{y}, \mathcal{G}, \mathcal{K}$)
2:     $n\_list = [0]_{\mathcal{K}}$
3:     $l\_list = [0]_{\mathcal{K} \times \mathcal{G}}, u\_list = [0]_{\mathcal{K} \times \mathcal{G}}$
4:     **for** client $k \in \mathcal{K}$ **in parallel do**
5:         $(l\_list[k], u\_list[k], n\_list[k])$
            $= \text{CLIENTCG}(k, \lambda, F_M, \tilde{y}, \mathcal{G})$
6:     **end for**
7:     $N = \sum_{k \in \mathcal{K}} n\_list[k], K = |\mathcal{K}|$
8:     $U_{\text{cov}} = [0]_{\mathcal{G}}, L_{\text{cov}} = [0]_{\mathcal{G}}$
9:     **for** client $k \in \mathcal{K}$ **do**
10:         $\gamma_k = ((n\_list[k] + 1)/(N + K))$
11:         $U_{\text{cov}} \mathrel{+}= \left(\gamma_k / L^{(g,\tilde{y})}\right) \cdot u\_list[k]$
12:         $L_{\text{cov}} \mathrel{+}= \left(\gamma_k / U^{(g,\tilde{y})}\right) \cdot l\_list[k]$
13:     **end for**
14:     $U_{\text{cov}} = \text{element\_wise\_min}(U_{\text{cov}}, [1]_{\mathcal{G}})$
15:     $\text{cov\_gap} = \max_{g \in \mathcal{G}} U_{\text{cov}}[g] - \min_{g \in \mathcal{G}} L_{\text{cov}}[g]$
16:     **return** cov\_gap
17: **end procedure**

**Algorithm 3** Client-Side Computation for Coverage Gap

1: **procedure** CLIENTCG($k, \lambda, F_M, \tilde{y}, \mathcal{G}$)
2:     $l_k = [0]_{\mathcal{G}}$
3:     $u_k = [0]_{\mathcal{G}}$
4:     **for** $g \in \mathcal{G}$ **do**
5:         **if** use_mle **then**
6:             $l_k[g] \leftarrow \frac{\alpha_k^{(g,\tilde{y});\lambda}}{n_k}$
7:             $u_k[g] \leftarrow \frac{\alpha_k^{(g,\tilde{y});\lambda}}{n_k}$
8:         **else**
9:             $l_k[g] \leftarrow \frac{\alpha_k^{(g,\tilde{y});\lambda} \cdot n_k^{(g,\tilde{y})}}{\left(n_k^{(g,\tilde{y})}+1\right) \cdot (n_k+1)}$
10:            $u_k[g] \leftarrow \frac{\alpha_k^{(g,\tilde{y});\lambda}+1}{(n_k+1)}$
11:         **end if**
12:     **end for**
13:     **return** $l_k, u_k, n_k$
14: **end procedure**

---

## 4 EXPERIMENTS

### 4.1 SETUP.

**Datasets.** We evaluate the FedCF framework on four multi-class datasets in different domains: (1, 2) ACSIncome and ACSEducation (Ding et al., 2021), (3) Pokec-{n, z} (Takac & Zabovsky, 2012), (4) Fitzpatrick Groh et al. (2021). These datasets were not originally for FL, so we partitioned them to form our clients. For the ACS datasets, we use state and territory data to partition the information

---

[2]The algorithm for computing the prior is similar to that of the coverage gap, so we omit it here

into clients. We consider **six** different partitioning schemes, based on common regional definitions in the U.S., which result in 4 (small), 8 (large), and 51 (all) clients. We also consider equivalent schemes for just the continental U.S.. For Pokec-{n, z}, each graph is treated as a separate client, as they originate from distinct partitions of the larger *Pokec* social network. Finally, for Fitzpatrick, since there is no predetermined partitioning scheme, we use a Dirichlet partitioner (Yurochkin et al., 2019) with concentration parameter of 0.5 to split $\mathcal{D}_{\text{train}}/\mathcal{D}_{\text{valid}}/\mathcal{D}_{\text{calib}}$ for $K \in \{2, 4, 8\}$ clients. We use a $30\%/20\%/25\%/25\%$ stratified split for the full $\mathcal{D}_{\text{train}}/\mathcal{D}_{\text{valid}}/\mathcal{D}_{\text{calib}}/\mathcal{D}_{\text{test}}$. We elaborate more on the datasets and experimental setup in Appendix C.

**Base Models.** For the ACS datasets, we use XGBoost (Chen & Guestrin, 2016). For Pokec-{n,z}, we use GraphSAGE (Hamilton et al., 2017) with GCN aggregation. For Fitzpatrick, we use ResNet-18 (He et al., 2016). Each of these models is trained using FedAvg (McMahan et al., 2017).

**Baseline.** We construct a federated fairness-agnostic conformal predictor targeting a coverage level of $1 - \alpha = 0.9$ using FCP Lu et al. (2023) with T-Digest Dunning (2021). For the non-conformity score, we adopt APS (Romano et al., 2020b) and RAPS (Angelopoulos et al., 2022) for all datasets, as well as DAPS (H. Zargarbashi et al., 2023), a graph-specific method, for Pokec-{n,z}. We then assess fairness using $\lambda = \hat{q}(\alpha)$ for three popular group-fairness metrics, reformulated in Table 2–Demographic Parity, Equal Opportunity, and Predictive Equality.

Table 2: Formulations for Conformal Fairness Metrics.

| Metric | Definition |
|---|---|
| Demographic (or Statistical) Parity | $\left\| \Pr\left[ \tilde{y} \in \mathcal{C}_\lambda(X) \mid X \in g_a \right] - \Pr\left[ \tilde{y} \in \mathcal{C}_\lambda(X) \mid X \in g_b \right] \right\| < c, \ \forall g_a, g_b \in \mathcal{G}, \ \forall \tilde{y} \in \mathcal{Y}^+$ |
| Equal Opportunity | $\left\| \Pr\left[ \tilde{y} \in \mathcal{C}_\lambda(X) \mid Y = \tilde{y}, X \in g_a \right] - \Pr\left[ \tilde{y} \in \mathcal{C}_\lambda(X) \mid Y = \tilde{y}, X \in g_b \right] \right\| < c, \ \forall g_a, g_b \in \mathcal{G}, \ \forall \tilde{y} \in \mathcal{Y}^+$ |
| Predictive Equality | $\left\| \Pr\left[ \tilde{y} \in \mathcal{C}_\lambda(X) \mid Y \neq \tilde{y}, X \in g_a \right] - \Pr\left[ \tilde{y} \in \mathcal{C}_\lambda(X) \mid Y \neq \tilde{y}, X \in g_b \right] \right\| < c, \ \forall g_a, g_b \in \mathcal{G}, \ \forall \tilde{y} \in \mathcal{Y}^+$ |

**Evaluation Metrics:** We report two key metrics: (1) *efficiency*, and (2) *worst-case fairness disparity*. The latter captures the largest difference in conditional coverage across groups, under the chosen fairness metric. For example, under *Demographic Parity*, we report:

$$\max_{\tilde{y} \in \mathcal{Y}^+} \max_{g_a, g_b \in \mathcal{G}} \left| \Pr\left[ \tilde{y} \in \mathcal{C}_\lambda(\boldsymbol{x}_{test}) \mid \boldsymbol{x}_{test} \in g_a \right] - \Pr\left[ \tilde{y} \in \mathcal{C}_\lambda(\boldsymbol{x}_{test}) \mid \boldsymbol{x}_{test} \in g_b \right] \right|. \tag{8}$$

More details on the experimental setup can be found in Appendix C.

## 4.2 RESULTS

In each figure, we use a **solid** line to represent the *average* efficiency of the **base federated conformal predictors** across different thresholds and a **dashed** line to represent the corresponding *average* worst-case fairness disparity. The bar plot shows the efficiency and worst-case fairness disparity using FedCF, while the **dots** indicate the *desired* fairness disparity. We report the average base performance for clarity and readability. In all experiments, FedCF achieves an actual fairness disparity within the specified closeness criterion, $c$, which may not be the case with the base federated conformal predictor.

**Preserves Key Characteristics of CF.** Two important characteristics of the CF framework are that it is (1) agnostic to the specific non-conformity score function and (2) supports intersectional fairness. We demonstrate that our FedCF framework preserves these two characteristics via the Pokec-{n, z} dataset. Pokec-{n, z} each have two sensitive attributes: *region* and *gender*. In addition to considering each attribute individually, we can treat each *pair* of attributes as distinct and apply FedCF. Furthermore, Pokec-{n, z} is a graph dataset. Recently, several developments have been made in graph CP research on non-conformity scores that utilize the graph structure. In addition to two standard CP methods–APS and RAPS–we also provide results using DAPS. Figure 1 shows how the FedCF framework can achieve the desired fairness criterion with minimal cost to efficiency for different non-conformity scores and when considering multiple groups.

**Robust Performance with Different Numbers of Clients.** An important trade-off in trustworthy FL is between predictive utility and maintaining fairness/privacy guarantees for each of its clients,

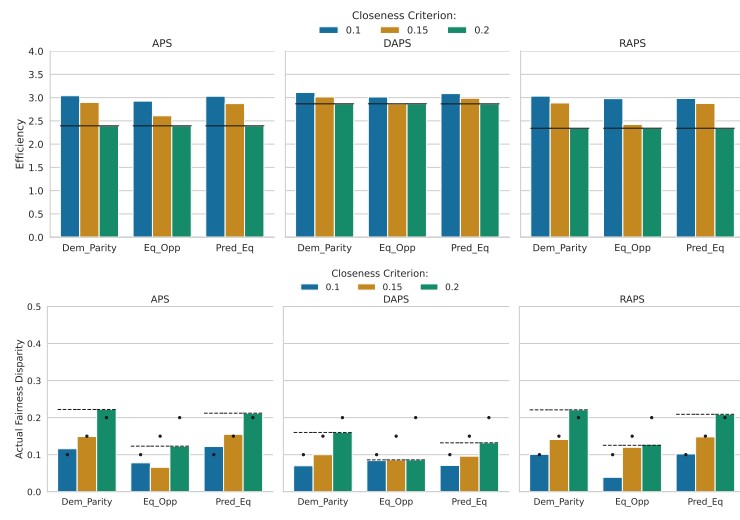

Figure 1: **Pokec-{n, z}** using **both** sensitive attributes. The top plots present the efficiency results, while the bottom plots are for the fairness disparities for (a) APS, (b) DAPS, and (c) RAPS. In all cases, FedCF achieves the desired closeness criteria better than the base federated conformal predictors.

which becomes increasingly challenging as the number of clients increases (Wen et al., 2023). To demonstrate how our framework can adapt to a varying number of clients, we use a Dirichlet partitioner with the Fitzpatrick dataset to evaluate performance with $K \in \{2, 4, 8\}$ clients in addition to a centralized setup with a single client. We see in Table 3 that as the number of clients increases, the baseline performance worsens, but the FedCF framework can still control for the necessary closeness criterion. We omit Equal Opportunity for Fitzpatrick as it is not meaningful in the context of this dataset, which aims to predict a skin condition, and the sensitive attribute is the skin type. People with certain skin types are known to be more likely to develop certain skin conditions, so the true positive rates of a classifier will typically not equalize, resulting in a degenerate (meaningless) solution. Finally, Fitzpatrick is a relatively small dataset with fewer than 17K points. Splitting the dataset into different splits and clients will result in a small number of points per $(g, \tilde{y})$ pair, making the interval bounds from Theorem 3.1 quite large.

Table 3: **Fitzpatrick using RAPS**. Each entry is of the form, **efficiency/fairness disparity**. We bold the lower fairness disparity value for each comparison. This table contains the results for $K \in \{1, 2, 4, 8\}$ clients. We observe that FedCF consistently matches or outperforms the base federated conformal predictor and is below the desired closeness criterion, $c$.

(a) $c = 0.1$

| Metric | 1 client Base | 1 client Ours | 2 clients Base | 2 clients Ours | 4 clients Base | 4 clients Ours | 8 clients Base | 8 clients Ours |
|---|---|---|---|---|---|---|---|---|
| **Dem_Parity** | 2.356 / 0.151 | 3.647 / **0.103** | 2.565 / 0.163 | 4.149 / **0.153** | 2.844 / 0.293 | 4.684 / **0.099** | 3.502 / 0.166 | 5.214 / **0.089** |
| **Pred_Eq** | 2.356 / 0.111 | 3.647 / **0.109** | 2.543 / 0.177 | 4.140 / **0.177** | 2.837 / 0.287 | 4.564 / **0.102** | 3.502 / 0.171 | 5.293 / **0.083** |

(b) $c = 0.15$

| Metric | 1 client Base | 1 client Ours | 2 clients Base | 2 clients Ours | 4 clients Base | 4 clients Ours | 8 clients Base | 8 clients Ours |
|---|---|---|---|---|---|---|---|---|
| **Dem_Parity** | 2.356 / **0.151** | 2.356 / **0.151** | 2.565 / 0.163 | 2.793 / **0.163** | 2.837 / 0.291 | 3.347 / **0.114** | 3.498 / 0.166 | 3.859 / **0.088** |
| **Pred_Eq** | 2.356 / **0.111** | 2.356 / **0.111** | 2.543 / 0.177 | 2.778 / **0.177** | 2.844 / 0.289 | 3.296 / **0.144** | 3.496 / 0.170 | 3.867 / **0.087** |

(c) $c = 0.2$

| Metric | 1 client Base | 1 client Ours | 2 clients Base | 2 clients Ours | 4 clients Base | 4 clients Ours | 8 clients Base | 8 clients Ours |
|---|---|---|---|---|---|---|---|---|
| **Dem_Parity** | 2.356 / **0.151** | 2.356 / **0.151** | 2.541 / 0.161 | 2.541 / **0.161** | 2.844 / 0.293 | 3.260 / **0.164** | 3.500 / 0.166 | 3.528 / **0.146** |
| **Pred_Eq** | 2.356 / **0.111** | 2.356 / **0.111** | 2.541 / 0.177 | 2.541 / **0.177** | 2.837 / 0.287 | 3.256 / **0.167** | 3.501 / 0.171 | 3.524 / **0.150** |

**Efficiency vs Fairness Trade-Off.** To make FedCF an actionable framework, it is essential to understand the utility trade-off when imposing fairness constraints. Using the interval-based approach to estimate the fairness-specific coverage gap gives a finite-sample guarantee for controlling fairness gaps. However, sometimes imposing fairness may result in a severe cost to utility. For example, a degenerate conformal predictor (one with near-full efficiency) is "fair," but completely impractical for use. By relaxing the theoretical guarantees, we can improve the efficiency by considering a tighter estimate for the coverage gap, using point estimates through MLE. Figure 2 compares the efficiency and fairness disparities when using the interval bounds vs the point estimate on the ACSEducation dataset. We observe that with the interval bounds, we always get within the closeness criterion, but the efficiencies are quite high. Alternatively, using point estimates may exceed the desired closeness criterion, but be more fair than the baseline and not sacrifice as much efficiency.

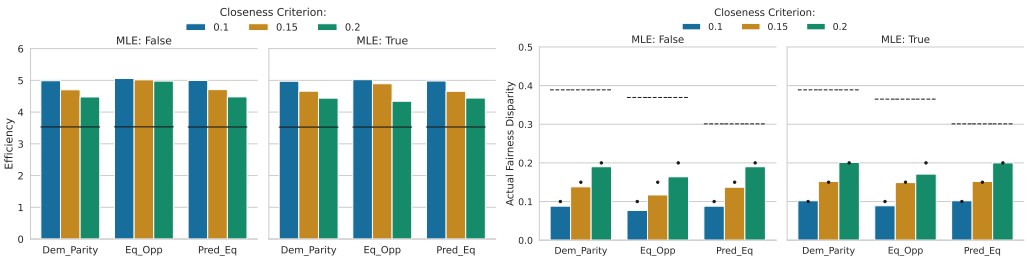

Figure 2: **ACSEducation using RAPS.** The left two plots are the efficiency plots for (a) using the interval bounds and (b) using the MLE estimate. Similarly, the two right (c) using the interval bounds and (d) using the MLE estimate. We observe that with the MLE estimates, FedCF achieves lower efficiency at the cost of a higher worst-case fairness disparity. Both the interval bounds and MLE estimates outperform the base federated conformal predictor in controlling for fairness disparity.

## 5 DISCUSSION

**On Data Heterogeneity.** In a practical federated setting, the data distribution will vary between clients–resulting in data heterogeneity across the FL system, which can affect performance at inference time (Wen et al., 2023). To address these concerns, we evaluate FedCF on varying partitioning schemes. For Fitzpatrick, we use a probabilistic partitioning scheme to ensure the data is distributed in a particular manner. For the ACS and Pokec-$\{n, z\}$ datasets, the partitioning is naturally induced by state and region information in the datasets.

**On Data Requirements.** A major limitation of CP is that to achieve a desired coverage rate, practically, you require a large enough calibration dataset such that the interval width for the CP guarantee is tight enough. This is exacerbated in the CF and FedCF framework as it requires sufficient calibration data for each group-positive label pair (for each client). If we consider intersectional fairness, the multiplicative increase in the number of groups further increases the data requirements.

**On Interval Bounds.** We provide two ways of estimating the fairness-specific coverage level with intervals and point estimates, but we can choose different intervals by considering the following. Suppose the event $s(\boldsymbol{x}_{test}, y_{test}) \leq \lambda$ (conditioned on $F_M$) is a Bernoulli random trial of some unknown probability $p$. We want to estimate $p$ as that is our fairness-specific coverage level. If we also treat the calibration scores as Bernoulli trials (by exchangeability), we can estimate $p$ using a Binomial Proportion Confidence Interval. Several results provide tighter or looser bounds (Wallis, 2013), which can be used to get better efficiency vs. fairness trade-offs.

## 6 CONCLUSION

In this work, we extended the Conformal Fairness framework to a federated setting, introducing the novel and comprehensive FedCF framework. We reformulated the CF framework to use a descent-based approach to make it more efficient for FL applications. Additionally, we developed theoretically grounded protocols to enable coverage gap calculations in a federated manner. The

FedCF framework offers clients a choice of participation protocols, including *communication efficient* and *enhanced privacy* options. We conducted experiments on various non-conformity scores and datasets– including graph data where we leverage the exchangeability assumption from CP.

**Extensibility and Future Work** An important application of FedCF is that it can be used to audit federated conformal predictors for fairness (discussed in Appendix G). In the future, we will explore how FedCF can be extended to split learning (Gupta & Raskar, 2018). Unlike federated Learning, which trains full models locally and aggregates updates, split learning divides the model across clients and server, sharing only partial computations (enhanced privacy, reduced compute).

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

## A  Notation Table

Table 4: Common notation used in FedCF.

| Notation | Defintion |
|---|---|
| $\mathcal{G}$ | The set of all demographic groups. |
| $\mathcal{Y}, \mathcal{Y}^+$ | The set of labels and positive/advantaged labels, respectively. |
| $g$ and $\tilde{y}$ | The group $g \in \mathcal{G}$ and $\tilde{y} \in \mathcal{Y}^+$ under consideration. |
| $F_M$ | Filter function for fairness metric $M$. |
| $c$ | Closeness criterion for a fairness specification. |
| $\lambda$ | Threshold used for constructing test prediction sets. |
| $\mathcal{K}$ | The set of clients, $\{1, \dots, K\}$. |
| $\mathcal{D}_{\text{train}}^{(k)}/\mathcal{D}_{\text{valid}}^{(k)}/\mathcal{D}_{\text{calib}}^{(k)}$ | Client $k$'s train/validation/calibration dataset. |
| $n_k$ and $N$ | $|\mathcal{D}_{\text{calib}}^{(k)}|$ and $\sum_{k=1}^{K} n_k$, respectively. |
| $\mathcal{S}_k^{(g,\tilde{y})}$ and $n_k^{(g,\tilde{y})}$ | $\left\{ (\boldsymbol{x}_i, y_i) \in \mathcal{D}_{\text{calib}}^{(k)} \mid F_M(\boldsymbol{x}_i, y_i, g, \tilde{y}) = 1 \right\}$ and $\left| \mathcal{S}_k^{(g,\tilde{y})} \right|$ |
| $\alpha_k^{(g,\tilde{y});\lambda}$ | $\sum_{(\boldsymbol{x}_i, \text{-}) \in \mathcal{S}_k^{(g,\tilde{y})}} \mathbf{1}[s(\boldsymbol{x}_i, \tilde{y}) \leq \lambda]$ |
| $E_k$ and $\gamma_k$ | The event $\boldsymbol{x}_{\text{test}}$ is exchangeable with data from client $k$ and $\Pr[E_k]$. |
| $L^{(g,\tilde{y})}, U^{(g,\tilde{y})}$ | Bounds for prior (term ⓘⓥ). |
| $\pi^{(g,\tilde{y})}$ | Point estimate for prior (term ⓘⓥ). |
| $L_{\text{cov}}, U_{\text{cov}}$ | Bounds for fairness-specific coverage level. |
| $\Pi_{\text{cov}}$ | Point estimate for fairness-specific coverage level. |

## B  Proofs

### B.1  Proof of Theorem 3.1

Recall, since the data is distributed across clients in the federated setting, we reformulated the fairness-specific coverage level as Equation 3. In doing so, the computation of the coverage level is split between the clients and the server. We present bounds and point estimates for each of the terms in Equation 3 across Lemmas B.1, B.2, and B.3, leading to a proof of Theorem 3.1.

#### B.1.1  Client-Side Estimates

Since each client operates independently with its own dataset, we can derive interval bounds for terms ⓘ and ⓘⓘ. For the point estimates approach, we use maximum likelihood estimators (MLEs) for each term, providing the tightest estimates.

**Lemma B.1.** *For each client $k$, group $g$, positive label $\tilde{y}$, and threshold $\lambda$, we get the following interval bounds:*

$$\frac{\alpha_k^{(g,\tilde{y});\lambda}}{n_k^{(g,\tilde{y})}+1} \leq \Pr[s(\boldsymbol{x}_{test}, \tilde{y}) \leq \lambda \mid F_M(\boldsymbol{x}_{test}, y_{test}, g, \tilde{y}) = 1, E_k] \leq \frac{\alpha_k^{(g,\tilde{y});\lambda}+1}{n_k^{(g,\tilde{y})}+1} \tag{9}$$

*If the data are IID, then we can use an MLE point estimate, given by the following:*

$$\Pr[s(\boldsymbol{x}_{test}, \tilde{y}) \leq \lambda \mid F_M(\boldsymbol{x}_{test}, y_{test}, g, \tilde{y}) = 1, E_k] = \frac{\alpha_k^{(g,\tilde{y});\lambda}}{n_k^{(g,\tilde{y})}} \tag{10}$$

The proof of Lemma B.1 is as follows:

*Proof.* We first observe that,

$$\Pr[s(\boldsymbol{x}_{\text{test}}, \tilde{y}) \leq \lambda \mid F_M(\boldsymbol{x}_{\text{test}}, y_{\text{test}}, g, \tilde{y}) = 1, \ E_k]$$
$$= \Pr_{\boldsymbol{x}_{\text{test}} \sim P_k}[s(\boldsymbol{x}_{\text{test}}, \tilde{y}) \leq \lambda \mid F_M(\boldsymbol{x}_{\text{test}}, y_{\text{test}}, g, \tilde{y}) = 1],$$

since exchangeability with the elements in $k$ is true iff $\boldsymbol{x}_{\text{test}}$ is sampled from $k$'s local distribution, $P_k$. The interval bounds follow from the conditional coverage guarantees given in CF (Vadlamani et al., 2025).

For the point estimate, we can model the event that the predicted score $s(\boldsymbol{x}_{test}, \tilde{y})$ falls below $\lambda$ as a Bernoulli random variable with success probability $p$. We can treat the $n_k^{(g, \tilde{y})}$ calibration points as individual Bernoulli trials, to then construct a maximum likelihood estimate (MLE) for $p$, which will be $\hat{p} = \frac{\alpha_k^{(g, \tilde{y}); \lambda}}{n_k^{(g, \tilde{y})}}$. $\qquad\square$

Lemma B.1 bounds the fair-conditional coverage for a particular group-label pair for the test covariate $(\boldsymbol{x}_{\text{test}}, y_{\text{test}})$. We next bound the coverage of the test covariate satisfying the Fairness Metric $(F_M)$, conditioned on the test point being exchangeable with data from client $k$ using Lemma B.2, and provide the proof below.

### B.1.2 Server-Side Estimates

Terms ⓘⓘⓘ and ⓘⓥ require a global view of the clients' data, so they are handled on the server.

For term ⓘⓘⓘ, we follow the setup by Lu et al. (2023), where given $n_k = \left| \mathcal{D}_{\text{calib}}^{(k)} \right|$, $\gamma_k := \Pr[E_k] \propto n_k + 1$ and $\sum_{k=1}^{K} \gamma_k = 1$. Finally, for term ⓘⓥ, we have that

$$\Pr[F_M(\boldsymbol{x}_{\text{test}}, y_{\text{test}}, g, \tilde{y}) = 1] = \sum_{k=1}^{K} \Pr[F_M(\boldsymbol{x}_{\text{test}}, y_{\text{test}}, g, \tilde{y}) = 1 \mid E_k] \cdot \Pr[E_k].$$

Using Lemma B.2, we can get an interval-bound and point-estimate as shown in the following lemma.

**Lemma B.2.** *For each client $k$, group $g$, and positive label $\tilde{y}$, we get the following interval bounds:*

$$\frac{n_k^{(g, \tilde{y})}}{n_k + 1} \leq \Pr[F_M(\boldsymbol{x}_{test}, y_{test}, g, \tilde{y}) = 1 \mid E_k] \leq \frac{n_k^{(g, \tilde{y})} + 1}{n_k + 1} \tag{11}$$

*If the data are IID, then we can use an MLE point estimate, given by the following:*

$$\Pr[F_M(\boldsymbol{x}_{test}, y_{test}, g, \tilde{y}) = 1 \mid E_k] = \frac{n_k^{(g, \tilde{y})}}{n_k} \tag{12}$$

*Proof.* To demonstrate the finite sample guarantee, we note that $F_M(\boldsymbol{x}, y, g, \tilde{y}) = 1$ for all $(\boldsymbol{x}, y) \in \mathcal{D}_{\text{calib}}^{(k)}$ are all exchangeable Bernoulli trials. Observe that conditioning on $E_k$ implies $\mathcal{D}_{\text{calib}+}^{(k)} := \mathcal{D}_{\text{calib}}^{(k)} \cup \{\boldsymbol{x}_{\text{test}}\}$ is an exchangeable sequence of length $n_k + 1$. Treating this as a finite 'bag' of covariates, we have

$$\forall_{(\boldsymbol{x}, y) \in \mathcal{D}_{\text{calib}+}^{(k)}}, \ \Pr[F_M(\boldsymbol{x}, y, g, \tilde{y}) = 1 \mid E_k] = \frac{\sum_{(\boldsymbol{x}_i, y_i) \in \mathcal{D}_{\text{calib}+}^{(k)}} F_M(\boldsymbol{x}_i, y_i, g, \tilde{y})}{n_k + 1}.$$

In other words, we have defined the probability of randomly selecting a covariate with $F_M(\boldsymbol{x}, y, g, \tilde{y}) = 1$. Since this applies to all points we know,

$$\Pr[F_M(\boldsymbol{x}_{\text{test}}, y_{\text{test}}, g, \tilde{y}) = 1 \mid E_k] = \frac{\sum_{(\boldsymbol{x}_i, y_i) \in \mathcal{D}_{\text{calib}+}^{(k)}} F_M(\boldsymbol{x}_i, y_i, g, \tilde{y})}{n_k + 1}. \tag{13}$$

Since we implicitly condition on $F_M(\boldsymbol{x}, y, g, \tilde{y}), \ \forall(\boldsymbol{x}, y) \in k$ (effectively making them deterministic), we can calculate the following bounds 13,

$$\frac{\sum_{(\boldsymbol{x}_i, y_i) \in \mathcal{D}_{\text{calib}}^{(k)}} F_M(\boldsymbol{x}_i, y_i, g, \tilde{y})}{n_k + 1} \leq \Pr[F_M(\boldsymbol{x}_{\text{test}}, y_{\text{test}}, g, \tilde{y}) = 1 \mid E_k]$$

$$\leq \frac{\sum_{(\boldsymbol{x}_i, y_i) \in \mathcal{D}_{\text{calib}}^{(k)}} F_M(\boldsymbol{x}_i, y_i, g, \tilde{y}) + 1}{n_k + 1}, \tag{14}$$

where the $+1$ term comes from the unknown value of $F_M(\boldsymbol{x}_{\text{test}}, y_{\text{test}}, g, \tilde{y})$. Substituting the sums with $n_k^{(g,\tilde{y})}$, proves the interval bounds.

For the point estimate, the event that $F_M = 1$ can be modeled as a Bernoulli random variable with success probability $p$. We can use the full $n_k$ calibration points as $n_k$ Bernoulli trials to construct an MLE for $p$, which will be $\hat{p} = \frac{n_k^{(g,\tilde{y})}}{n_k}$. $\qquad\qquad\square$

Lastly, we use Lemma B.3, to bound $\Pr[F_M(\boldsymbol{x}_{\text{test}}, y_{\text{test}}, g, \tilde{y}) = 1]$. The proof of Lemma B.3 leverages the result of Lemma B.2.

**Lemma B.3.** *For each client $k$, group $g$, and positive label $\tilde{y}$, we get the following interval bounds:*

$$L^{(g,\tilde{y})} = \sum_{k=1}^{K} \gamma_k \frac{n_k^{(g,\tilde{y})}}{n_k+1} \leq \Pr[F_M(\boldsymbol{x}_{test}, y_{test}, g, \tilde{y}) = 1] \leq \sum_{k=1}^{K} \gamma_k \frac{n_k^{(g,\tilde{y})}+1}{n_k+1} = U^{(g,\tilde{y})}. \quad (15)$$

*If the data are IID, then we can use an MLE point estimate, given by the following:*

$$\Pr[F_M(\boldsymbol{x}_{test}, y_{test}, g, \tilde{y}) = 1] = \sum_{k=1}^{K} \gamma_k \frac{n_k^{(g,\tilde{y})}}{n_k} = \pi^{(g,\tilde{y})}. \quad (16)$$

*Proof.* To achieve this result, we first use the law of total probability to separate $\Pr[F_M(\boldsymbol{x}_{\text{test}}, y_{\text{test}}, g, \tilde{y})]$ into terms known by the server and the client:

$$\Pr[F_M(\boldsymbol{x}_{\text{test}}, y_{\text{test}}, g, \tilde{y})] = \sum_{k=1}^{K} \Pr[F_M(\boldsymbol{x}_{\text{test}}, y_{\text{test}}, g, \tilde{y}) = 1 \mid E_k] \cdot \Pr[E_k] \quad (17)$$

Then, substituting the bounds for term Ⅱ (see Lemma B.2), and $\gamma_k = \Pr[E_k]$ from term Ⅳ into Equation 17, we complete the proof. $\qquad\qquad\square$

Having proved the Lemmas, we can move on to proving,

**Theorem 3.1.** *The fairness-specific coverage level (Equation 3) can be bounded as*

$$L_{cov}(\lambda, F_M, g, \tilde{y}) \leq \Pr[s(\boldsymbol{x}_{test}, \tilde{y}) \leq \lambda \mid F_M(\boldsymbol{x}_{test}, y_{test}, g, \tilde{y}) = 1] \leq U_{cov}(\lambda, F_M, g, \tilde{y}),$$

*where*

$$L_{cov}(\lambda, F_M, g, \tilde{y}) = \sum_{k=1}^{K} \frac{\gamma_k \alpha_k^{(g,\tilde{y});\lambda} n_k^{(g,\tilde{y})}}{(n_k^{(g,\tilde{y})}+1)(n_k+1)U^{(g,\tilde{y})}} \text{ and } U_{cov}(\lambda, F_M, g, \tilde{y}) = \sum_{k=1}^{K} \frac{\gamma_k(\alpha_k^{(g,\tilde{y});\lambda}+1)}{(n_k+1)L^{(g,\tilde{y})}}. \quad (4)$$

*If the data is IID, using MLE estimates for each term, we get the following estimate for the fairness-specific coverage level*

$$\Pi_{cov} = \Pr[s(\boldsymbol{x}_{test}, \tilde{y}) \leq \lambda \mid F_M(\boldsymbol{x}_{test}, y_{test}, g, \tilde{y}) = 1] = \sum_{k=1}^{K} \frac{\gamma_k \alpha_k^{(g,\tilde{y});\lambda}}{n_k \pi^{(g,\tilde{y})}}. \quad (5)$$

*Proof.* Substituting the bounds for terms Ⅰ, Ⅱ, and Ⅳ which were established via Lemmas B.1, B.2, B.3 respectively and the defintion of Ⅲ into Equation 3 completes the proof. $\qquad\square$

On closer inspection, we observe that Terms Ⅰ and Ⅱ can be combined and bound together.

**Lemma B.4.** *Using the defintions of $\alpha_k^{(g,\tilde{y});\lambda}$ and $n_k + 1$ we have,*

$$\frac{\alpha_k^{(g,\tilde{y});\lambda}}{n_k+1} \leq \Pr\Big[s(\boldsymbol{x}_{test}, \tilde{y}) \leq \lambda \mid F_M(\boldsymbol{x}_{test}, y_{test}, g, \tilde{y}) = 1, \ \boldsymbol{x}_{test} \overset{exc.}{\sim} k\Big]$$

$$\cdot \Pr\Big[F_M(\boldsymbol{x}_{test}, y_{test}, g, \tilde{y}) = 1 \mid \boldsymbol{x}_{test} \overset{exc.}{\sim} k\Big] \leq \frac{\alpha_k^{(g,\tilde{y});\lambda}+1}{n_k+1}. \quad (18)$$

*Proof.* First, observe that,

$$\Pr[s(\boldsymbol{x}_{\text{test}}, \tilde{y}) \leq \lambda \mid F_M(\boldsymbol{x}_{\text{test}}, y_{\text{test}}, g, \tilde{y}) = 1, \ E_k] \cdot \Pr[F_M(\boldsymbol{x}_{\text{test}}, y_{\text{test}}, g, \tilde{y}) = 1 \mid E_k]$$

$$= \Pr[s(\boldsymbol{x}_{\text{test}}, \tilde{y}) \leq \lambda, \ F_M(\boldsymbol{x}_{\text{test}}, y_{\text{test}}, g, \tilde{y}) = 1 \mid E_k]$$

Then consider the following Bernoulli random variables (R.V) $\mathbf{1}[s(\boldsymbol{x}, y) \leq \lambda] \cdot F_M(\boldsymbol{x}, y, g, \tilde{y})$ for all $(\boldsymbol{x}, y) \in k \cup \{(\boldsymbol{x}_{\text{test}}, y_{\text{test}})\}$ which form an exchangeable sequence (using the assumption $\boldsymbol{x}_{\text{test}} \overset{\text{exc.}}{\sim} k$). Additionally, observe $\alpha_k^{(g,\tilde{y});\lambda} = \sum_{(\boldsymbol{x}_i, y_i) \in k} \mathbf{1}[s(\boldsymbol{x}, y) \leq \lambda] \cdot F_M(\boldsymbol{x}, y, g, \tilde{y})$ is an equivalent definition of $\alpha_k^{(g,\tilde{y});\lambda}$. The rest of the proof follows from the proof of Lemma B.2 by using $\mathbf{1}[s(\boldsymbol{x}, y) \leq \lambda] \cdot F_M(\boldsymbol{x}, y, g, \tilde{y})$ as the Bernoulli R.V instead of $F_M(\boldsymbol{x}, y, g, \tilde{y})$ and $\alpha_k^{(g,\tilde{y});\lambda}$ in place of $n_k^{(g,\tilde{y})}$. $\qquad\square$

Using the above result, we can tighten the lower-bound of Theorem 3.1.

**Corollary B.1.** *Swapping terms ① and ② with the combined term in Lemma B.4, the lower bound in Theorem 3.1 can be simplified and tightened to*

$$L_{cov}(\lambda, F_M, g, \tilde{y}) = \sum_{k=1}^{K} \frac{\gamma_k \alpha_k^{(g,\tilde{y});\lambda}}{(n_k + 1)U^{(g,\tilde{y})}} \tag{19}$$

*Proof.* Instead of substituting terms ① and ② into Equation 3 as in the proof of Theorem 3.1, we can instead use Lemma B.4 to update the bounds. Observe that the upper bound remains the same as Theorem 3.1 while the lower bound becomes tighter. $\qquad\square$

## C  ADDITIONAL EXPERIMENT DETAILS

### C.1  DATASETS

We present a summary of common dataset statistics in Table 5 and go into more details on each dataset in the following sections.

Table 5: Dataset Statistics. T refers to Tabular, G refers to Graph, and V refers to vision.
*ACS datasets have six (6) groups if using the *continental* split schemes (see Section C.2).
^ Number of inputs after removing those with unknown group information

| Name | Type | Size | # Labeled | # Groups | # Classes |
|------|------|------|-----------|----------|-----------|
| ACSIncome | T | $1,664,500$ | ALL | race$(9)^*$ | 4 |
| ACSEducation | T | $1,664,500$ | ALL | race$(9)^*$ | 6 |
| Fitzpatrick | V | $16,012$^ | ALL | skin type$(6)$ | 9 |

| Name | Type | $(|\mathcal{V}|, |\mathcal{E}|)$ | # Labeled | # Groups | # Classes |
|------|------|------|-----------|----------|-----------|
| Pokec-{n, z} | G | $(133,138, 1,458,258)$ | $17,594$ | region$(2)$, gender$(2)$ | 4 |

### C.2  FOLKTABLES DATASETS

In the fairness space, the American Community Services (ACS) datasets from the `Folktables` library are a widely used set of tabular data (Ding et al., 2021). The data is taken across the 51 U.S. states and territories. For our federated setup and each dataset below, we consider the following 6 partitioning schemes:

(1.) **All:** We consider each U.S. state and territory to be its own client

(2.) **Large:** We follow the U.S. Census Bureau's division of the U.S. into the Northeast, the Midwest, the South, and the West

(3.) **Small:** We follow the Bureau of Economic Analysis's division of the U.S. into New England, the Mideast, the Great Lakes, the Plains, the Southeast, the Southwest, the Rocky Mountain, and the Far West.

(4-6.) **Continental All, Continental Large, Continental Small:** The same as 1 to 3, but we only consider the *continental* U.S.–removing Alaska, Hawaii, and Puerto Rico.

All Folktable datasets have a race attribute. When we partition the data using all the states and territories, we use the full version of race, which has 9 groups. However, when partitioning just with *continental* U.S., we combine some demographic groups–primarily those from Alaska, Hawaii, and Puerto Rico–into the appropriate 'Other' categories, resulting in a total of 6 groups.

**ACSIncome:** We used the standard ACSIncome dataset from Folktables; however, we divided the targets into four classes by evenly splitting the income into 4 brackets. The sensitive attribute in this case is race, resulting in either 9 or 6 groups.

**ACSEducation:** This is a custom dataset. We used the ACSTravelTime data and selected Education Level as the target. The education level was divided into 6 groups: {did not complete high school, has a high school diploma, has a GED, started an undergrad program, completed an undergrad program, and completed graduate or professional school}. ACSEducation also uses race as a sensitive attribute.

### C.3 NON-TABULAR DATASETS

**Pokec-{n,z}:** The Pokec-{n, z} dataset (Takac & Zabovsky, 2012) is a social network graph dataset collected from Pokec, a popular social network in Slovakia. Since several rows in the dataset are missing features, two commonly used subgraphs are the Pokec-z and Pokec-n datasets. The graphs have four labels corresponding to the fieldwork and two sensitive attributes: gender (2 groups) and region (2 groups). Our experiments consider each attribute individually as well as intersectional fairness by creating an attribute with 4 groups. For our federated setup, we use each subgraph as a single client, resulting in 2 clients.

**Fitzpatrick:** The Fitzpatrick dataset (Groh et al., 2021) contains clinical images classified based on the depicted skin condition. There are several levels of granularity regarding the skin condition label. We use a version with 9 skin conditions: {inflammatory, malignant epidermal, genodermatoses, benign dermal, benign epidermal, malignant melanoma, benign melanocyte, malignant cutaneous lymphoma, malignant dermal}. There are 6 demographic groups based on the Fitzpatrick skin type. For our federated setup, we use a Dirichlet partitioner to split the data into $K \in \{2, 4, 8\}$ clients.

### C.4 HYPERPARAMETERS AND IMPLEMENTATION

To promote reproducibility, the source code for FedCF is provided in the supplementary material, along with the configuration files containing the hyperparameters used.

The project was written using the Flower AI Federated Learning framework (Beutel et al., 2020) for both base model training and the FedCF framework.

### C.5 NON-CONFORMITY SCORES

**Adaptive Prediction Sets (APS)** The most popular CP method for classification problems is APS (Romano et al., 2020b). The scoring function first sorts the softmax logits in descending order and accumulates the class probabilities until the correct class is included. For tighter prediction sets, randomization is introduced through a uniform random variable.

Formally, let $\hat{\pi}$ be a trained classification model with softmaxed output. If $\hat{\pi}(\boldsymbol{x})_{(1)} \geq \hat{\pi}(\boldsymbol{x})_{(2)} \geq \cdots \geq \hat{\pi}(\boldsymbol{x})_{(K-1)}$, $u \sim U(0, 1)$, and $r_y$ is the rank of the correct label, then

$$s(\boldsymbol{x}, y) = \left[ \sum_{i=1}^{r_y} \hat{\pi}(\boldsymbol{x})_{(i)} \right] - u\hat{\pi}(\boldsymbol{x})_y.$$

APS has two major drawbacks that have led to it being surpassed by other methods in recent CP literature. First, APS tends to produce large (less efficient) prediction sets. Second, it does not

account for structure in its formulation. To address these issues, alternatives like RAPS and DAPS have emerged[3].

**Regularized Adaptive Prediction Sets (RAPS)**  Angelopoulos et al. (2022) introduces a regularization approach for APS. Given the same setup and notation as APS, define $o(\boldsymbol{x}, y) = |\{c \in \mathcal{Y} : \hat{\pi}(\boldsymbol{x})_y \geq \hat{\pi}(\boldsymbol{x})_c\}|$. Then,

$$s(\boldsymbol{x}, y) = \left[\sum_{i=1}^{r_y} \hat{\pi}(\boldsymbol{x})_{(i)}\right] - u\hat{\pi}(\boldsymbol{x})_y + \nu \cdot \max\{(o(\boldsymbol{x}, y) - k_{reg}), 0\},$$

where $\nu$ and $k_{reg} \geq 0$ are regularization hyperparameters.

**Diffusion Adaptive Prediction Sets (DAPS)**  Graphs are rich with neighborhood information, with nodes often exhibiting homophily. This suggests that the non-conformity scores of connected nodes are likely to be related. To leverage this insight, DAPS H. Zargarbashi et al. (2023) incorporates a one-step diffusion update on the non-conformity scores. Formally, if $s(\boldsymbol{x}, y)$ is a point-wise score function (e.g., APS), then the diffusion step yields a new score function

$$\hat{s}(\boldsymbol{x}, y) = (1 - \delta)s(\boldsymbol{x}, y) + \frac{\delta}{|\mathcal{N}_{\boldsymbol{x}}|} \sum_{\boldsymbol{u} \in \mathcal{N}_{\boldsymbol{x}}} s(\boldsymbol{u}, y),$$

where $\delta \in [0, 1]$ is a diffusion hyperparamter and $\mathcal{N}_{\boldsymbol{x}}$ is the 1-hop neighborhood of $\boldsymbol{x}$.

# D  FedCF with Enhanced Privacy

Preserving data privacy is a fundamental pillar of FL mechanisms, as they typically interact with sensitive client data. In this vein, we formulate an *enhanced privacy* version of FedCF.

## D.1  Enhanced Privacy

To better preserve privacy (compared to the *communication efficient* approach), we can offload more of the computation to the client-side, making it harder for the server-side to reverse-engineer or infer distributional information from the sent quantities. Expanding Equation 7, we get

$$U_{\text{cov}}(\lambda, F_m, g_a, \tilde{y}) - L_{\text{cov}}(\lambda, F_m, g_b.\tilde{y})$$
$$= \sum_{k=1}^{K} \gamma_k \underbrace{\left\{\frac{(\alpha_k^{(g_a, \tilde{y}); \lambda} + 1)}{(n_k + 1)L^{(g_a, \tilde{y})}} - \frac{\alpha_k^{(g_b, \tilde{y}); \lambda} n_k^{(g_b, \tilde{y})}}{(n_k^{(g_b, \tilde{y})} + 1)(n_k + 1)U^{(g_b, \tilde{y})}}\right\}}_{\text{Returned by the Client}}. \tag{20}$$

In this formulation, the client sends back the summand for each group, positive label pair, making the space complexity of the client's message $\mathcal{O}(|\mathcal{G}|^2|\mathcal{Y}^+|)$–quadratic with respect to the number of groups and linear with respect to positive labels.

The data privacy improves with this approach compared to the *communication efficient* version, since the data sent to the server is the difference of client-level summary statistics, which obfuscates individual distribution information from the server. However, unlike the *communication efficient* approach, the upper-coverage term ($U_{\text{cov}}$) is not separable from the aggregated sum, thus preventing us from enforcing $U_{\text{cov}} < 1$. In limited data settings, this results in more conservative coverage gap estimates, which increases the prediction set size when using the *enhanced privacy* approach.

We provide a side-by-side comparison of the *communication efficient* and *enhanced privacy* version of computing the federated coverage gap in Figure 3 in Appendix E.

---

[3]RAPS and DAPS have hyperparameters typically tuned on separate held-out data, but we fix them *a priori* to preserve data for calibration and evaluation as well as to be consistent with what prior federated conformal prediction works have done.

## D.2 Hybrid

In real-world scenarios, clients often have varying privacy and communication requirements. For example, clients in resource-constrained areas may not have the network bandwidth to send the necessary packets to the centralized server. In our proposed *hybrid* approach, a client may elect to be *communication efficient*, without preventing the remaining clients from using the *enhanced privacy* protocol. We present the full server-side algorithm, which combines the *communication efficient*, *enhanced privacy*, and *hybrid* protocols for the federated coverage gap, in Algorithm 6 in Appendix E.

## D.3 Empirical Comparison

We conduct two experiments using the Fitzpatrick dataset and 8 clients, as well as the larger AC-SIncome dataset with the *continental_all* partition scheme–48 clients–to test the *communication efficient*, *enhanced privacy*, and *hybrid* protocols. For the *hybrid* protocol, we randomly assign half the clients to each protocol. From Table 6, we observe that all configurations control the fairness disparity within the closeness criterion; however, if all clients agree upon the *communication efficient* protocol, FedCF achieves a better efficiency with a slightly worse fairness disparity, albeit still within the closeness criterion. Though with more data, we observe that the efficiency gaps are smaller as seen in Table 7.

Table 6: **Fitzpatrick, 8 clients, APS.** Each entry is of the form, **efficiency/fairness disparity**. We bold the lower fairness disparity value for each comparison. We observe that the *communication efficient* approach produces the most efficient prediction sets, while having a similar or higher fairness disparity. The *enhanced privacy* approach and *hybrid* approach have similar performance (w.r.t efficiency and fairness disparity), with minor differences stemming from the stochasticity of FedCF, as they default to the same coverage-gap aggregation protocol (see Algorithm 6). All methods improve upon the baseline fairness disparity and control for the closeness criterion.

(a) Enhanced Privacy

| Metric | $c = 0.1$ | | $c = 0.15$ | | $c = 0.2$ | |
|---|---|---|---|---|---|---|
| | Base | Ours | Base | Ours | Base | Ours |
| **Dem_Parity** | 3.671 / 0.136 | 7.041 / **0.047** | 3.671 / 0.136 | 4.978 / **0.101** | 3.672 / 0.136 | 3.940 / **0.111** |
| **Pred_Eq** | 3.676 / 0.134 | 7.042 / **0.047** | 3.675 / 0.134 | 4.765 / **0.094** | 3.672 / 0.134 | 3.8803 / **0.106** |

(b) Hybrid (50-50)

| Metric | $c = 0.1$ | | $c = 0.15$ | | $c = 0.2$ | |
|---|---|---|---|---|---|---|
| | Base | Ours | Base | Ours | Base | Ours |
| **Dem_Parity** | 3.674 / 0.136 | 6.871 / **0.066** | 3.670 / 0.136 | 4.967 / **0.103** | 3.670 / 0.136 | 3.939 / **0.111** |
| **Pred_Eq** | 3.671 / 0.134 | 7.041 / **0.047** | 3.673 / 0.134 | 5.123 / **0.094** | 3.671 / 0.134 | 3.919 / **0.107** |

(c) Communication Efficient

| Metric | $c = 0.1$ | | $c = 0.15$ | | $c = 0.2$ | |
|---|---|---|---|---|---|---|
| | Base | Ours | Base | Ours | Base | Ours |
| **Dem_Parity** | 3.674 / 0.137 | 6.053 / **0.104** | 3.674 / 0.136 | 4.890 / **0.103** | 3.672 / 0.136 | 3.935 / **0.111** |
| **Pred_Eq** | 3.671 / 0.134 | 6.308 / **0.109** | 3.674 / 0.134 | 4.931 / **0.094** | 3.674 / 0.134 | 3.876 / **0.106** |

Table 7: **ACSIncome, Continental All, RAPS.** Each entry is of the form, **efficiency/fairness disparity**. We observe that with sufficient data, each protocol performs at a similar efficiency, and they all decrease the baseline fairness disparity and control it within the closeness criterion. Our fairness disparity values are bolded.

(a) Enhanced Privacy

| Metric | $c = 0.1$ | | $c = 0.15$ | | $c = 0.2$ | |
|---|---|---|---|---|---|---|
| | Base | Ours | Base | Ours | Base | Ours |
| **Dem_Parity** | 2.609 / 0.148 | 3.037 / **0.086** | 2.610 / 0.148 | 2.634 / **0.138** | 2.613 / 0.148 | 2.613 / **0.148** |
| **Pred_Eq** | 2.607 / 0.160 | 3.294 / **0.063** | 2.610 / 0.161 | 2.661 / **0.138** | 2.609 / 0.161 | 2.609 / **0.161** |

(b) Hybrid (50-50)

| Metric | $c = 0.1$ | | $c = 0.15$ | | $c = 0.2$ | |
|---|---|---|---|---|---|---|
| | Base | Ours | Base | Ours | Base | Ours |
| **Dem_Parity** | 2.608 / 0.148 | 3.039 / **0.085** | 2.609 / 0.148 | 2.633 / **0.138** | 2.596 / 0.148 | 2.596 / **0.148** |
| **Pred_Eq** | 2.606 / 0.160 | 3.277 / **0.079** | 2.606 / 0.160 | 2.657 / **0.138** | 2.595 / 0.161 | 2.595 / **0.161** |

(c) Communication Efficient

| Metric | $c = 0.1$ | | $c = 0.15$ | | $c = 0.2$ | |
|---|---|---|---|---|---|---|
| | Base | Ours | Base | Ours | Base | Ours |
| **Dem_Parity** | 2.608 / 0.148 | 3.037 / **0.086** | 2.611 / 0.148 | 2.634 / **0.138** | 2.601 / 0.149 | 2.601 / **0.149** |
| **Pred_Eq** | 2.610 / 0.161 | 3.300 / **0.071** | 2.607 / 0.160 | 2.658 / **0.138** | 2.609 / 0.161 | 2.609 / **0.161** |

With these empirical results, note that under the hybrid setting, clients that optimize for communication efficiency still benefit from the fact that they can operate over a limited bandwidth network connection. The required bandwidth for a particular client undergoes a factor of $\approx \frac{|\mathcal{G}|}{2}$ reduction– i.e. $\mathcal{O}(|\mathcal{G}^2||\mathcal{Y}^+) \to \mathcal{O}(2 \cdot |\mathcal{G}||\mathcal{Y}^+)$, when a client selects the *communication efficient* protocol while ensuring the remaining clients benefit from the *enhanced privacy* protocol. For Fitzpatrick, this results in the communication overhead (in bytes) being reduced by a factor of three. ($|\mathcal{G}|/2 = 3$, for Fitzpatrick). For the ACS datasets using the small, large, or all client assignments, this reduction corresponds to $|\mathcal{G}|/2 = 4.5$

# E  ALGORITHMS

**Algorithm 4** More Communication Efficient Client-Side Computation for Coverage Gap

1: **procedure** CLIENTCG_COMM_EFFICIENT($k$, $\lambda$, $F_M$, $\tilde{y}$, $\mathcal{G}$)
2:     $l_k = [0]_{\mathcal{G}}$
3:     $u_k = [0]_{\mathcal{G}}$
4:     **for** $g \in \mathcal{G}$ **do**
5:         **if** use_mle **then**
6:             $l_k[g] \leftarrow \dfrac{\alpha_k^{(g,\tilde{y});\lambda}}{n_k}$
7:             $u_k[g] \leftarrow \dfrac{\alpha_k^{(g,\tilde{y});\lambda}}{n_k}$
8:         **else**
9:             $l_k[g] \leftarrow \dfrac{\alpha_k^{(g,\tilde{y});\lambda} \cdot n_k^{(g,\tilde{y})}}{\left(\left(n_k^{(g,\tilde{y})}+1\right) \cdot (n_k+1)\right)}$
10:            $u_k[g] \leftarrow \dfrac{\alpha_k^{(g,\tilde{y});\lambda}+1}{(n_k+1)}$
11:        **end if**
12:    **end for**
13:    **return** $l_k, u_k, n_k$
14: **end procedure**

**Algorithm 5** Client-Side Computation for Coverage Gap with Enhanced Privacy

1: **procedure** CLIENTCG_PRIVATE($k$, $\lambda$, $F_M$, $\tilde{y}$, $\mathcal{G}$)
2:     $l_k = [0]_{\mathcal{G}}$
3:     $u_k = [0]_{\mathcal{G}}$
4:     **for** $g \in \mathcal{G}$ **do**
5:         **if** use_mle **then**
6:             $l_k[g] \leftarrow \dfrac{\alpha_k^{(g,\tilde{y});\lambda}}{\left(n_k \cdot \pi^{(g,\tilde{y})}\right)}$
7:             $u_k[g] \leftarrow \dfrac{\alpha_k^{(g,\tilde{y});\lambda}}{\left(n_k \cdot \pi^{(g,\tilde{y})}\right)}$
8:         **else**
9:             $l_k[g] \leftarrow \dfrac{\alpha_k^{(g,\tilde{y});\lambda} \cdot n_k^{(g,\tilde{y})}}{\left(\left(n_k^{(g,\tilde{y})}+1\right) \cdot (n_k+1) \cdot U^{(g,\tilde{y})}\right)}$
10:            $u_k[g] \leftarrow \dfrac{\alpha_k^{(g,\tilde{y});\lambda}+1}{\left((n_k+1) \cdot L^{(g,\tilde{y})}\right)}$
11:        **end if**
12:    **end for**
13:    $pw\_cg_k = [0]_{\mathcal{G} \times \mathcal{G}}$
14:    // Pairwise coverage gap
15:    **for** $(g_a, g_b) \in \mathcal{G} \times \mathcal{G}$ **do**
16:        $pw\_cg_k[g_a, g_b] \leftarrow u_k[g_a] - l_k[g_b]$
17:    **end for**
18:    **return** $pw\_cg_k, n_k$
19: **end procedure**

Figure 3: **Pseudocode for the two client-side protocols to compute the coverage gap.** The *enhanced privacy* version (on the right) includes the pairwise computation step, which results in a larger space complexity compared to the more *communication efficient* version (on the left).

---

**Algorithm 6** Full Server-side Aggregation for Coverage Gap

---

1: **procedure** SERVERCG($\lambda_0, F_M, \tilde{y}, \mathcal{G}$, formulations)
2:     $n\_list = [0]_\mathcal{K}$
3:     $l\_list = [0]_{\mathcal{K} \times \mathcal{G}}, u\_list = [0]_{\mathcal{K} \times \mathcal{G}}$               ▷ Used for comm. efficient formulations
4:     $pw\_cg\_list = [0]_{\mathcal{K} \times \mathcal{G} \times \mathcal{G}}$                       ▷ Used for private formulations
5:     **for** client $k \in \mathcal{K}$ **in parallel do**
6:         **if** formulations $==$ COMM_EFFICIENT **then**
7:             Receive $(l_k, u_k, n_k) =$ CLIENTCG_COMM_EFFICIENT$(k, \lambda_0, F_M, \tilde{y}, \mathcal{G})$
8:             $l\_list[k] \leftarrow l_k, u\_list[k] \leftarrow u_k$
9:         **else**
10:            Receive $(pw\_cg_k, n_k) =$ CLIENTCG_PRIVATE$(k, \lambda_0, F_M, \tilde{y}, \mathcal{G})$
11:            $pw\_cg\_list[k] \leftarrow pw\_cg_k$
12:         **end if**
13:         $n\_list[k] \leftarrow n_k$
14:     **end for**

15:     // Initialize final coverage variables
16:     $N = \sum_{k \in \mathcal{K}} n\_list[k], K = |\mathcal{K}|, U_{\text{cov}} = [0]_\mathcal{G}, L_{\text{cov}} = [0]_\mathcal{G}, PW_{\text{cov}} = [0]_{\mathcal{G} \times \mathcal{G}}$
17:     all_comm_efficient $=$ all(formulations$[k] ==$ COMM_EFFICIENT)
18:     **for** client $k \in \mathcal{K}$ **do**
19:         $\gamma_k = ((n\_list[k] + 1)/(N + K))$
20:         **if** all_comm_efficient **then**
21:            $U_{\text{cov}}$ += $\left( \gamma_k / L^{(g, \tilde{y})} \right) \cdot u\_list[k]$          ▷ Standard operations are element-wise
22:            $L_{\text{cov}}$ += $\left( \gamma_k / U^{(g, \tilde{y})} \right) \cdot l\_list[k]$
23:         **else**
24:            **if** formulations$[k] ==$ COMM_EFFICIENT **then**
25:                $PW_{\text{cov}}$ += $\gamma_k \cdot \left( u\_list[k] \ominus l\_list[k]^\top \right)$ ▷ $\ominus$ is pairwise differences between two vectors.
26:            **else**
27:                $PW_{\text{cov}}$ += $\gamma_k \cdot pw\_cg\_list[k]$
28:            **end if**
29:         **end if**
30:     **end for**

31:     **if** all_comm_efficient **then**
32:         $U_{\text{cov}} = $ element_wise_min$(U_{\text{cov}}, [1]_\mathcal{G})$     ▷ Limit upper coverage prior to coverage gap calculation
33:         cov_gap $= \max_{g \in \mathcal{G}} U_{\text{cov}}[g] - \min_{g \in \mathcal{G}} L_{\text{cov}}[g]$
34:     **else**
35:         cov_gap $= \min \left\{ \max_{g_a, g_b \in \mathcal{G}} PW_{\text{cov}}[g_a, g_b], 1 \right\}$         ▷ Limit Coverage Gap to 1
36:     **end if**
37:     **return** cov_gap
38: **end procedure**

---

## F    DIFFERENTIAL PRIVACY IN FEDCF

FedCF can also be extended to formally consider $(\epsilon, \delta)$-differential privacy (DP), a mathematically rigorous framework for data privacy (Dwork, 2006), where $\delta$ is the probability that $\epsilon$-DP is violated. We can embed DP within our framework via client shuffling and additive noise approaches. Client shuffling is a global DP approach that is performed after the client sends data. Before the server receives the data, it goes through a trusted, centralized shuffler to anonymize which client has sent what data (Erlingsson et al., 2019). Our framework can accommodate client shuffling due to its parallelism with client-side computation and its additive aggregation approach.

For additive noise, we propose augmenting the values each client sends back with Gaussian noise (Dwork et al., 2014; Dong et al., 2022), such that a client returns,

$$h = \frac{(\alpha_k^{(g_a,\tilde{y});\lambda} + 1)}{(n_k + 1)L^{(g_a,\tilde{y})}} - \frac{\alpha_k^{(g_b,\tilde{y});\lambda} n_k^{(g_b,\tilde{y})}}{(n_k^{(g_b,\tilde{y})} + 1)(n_k + 1)U^{(g_b,\tilde{y})}} + X, \tag{21}$$

where $X$ is a Gaussian random variable (R.V). For the *communication efficient* approach, one would add a Gaussian R.V. to the upper coverage and lower coverage terms returned by the client. To ensure $(\epsilon, \delta)$-DP, we make $X \sim \mathcal{N}\left(0, \frac{2\ln(1.25/\delta)(\Delta g)^2}{\epsilon^2}\right)$, where $\Delta h$ is the sensitivty of $h$–or how much $h$ can change if one of the points in the client's dataset changes. For FedCF, $h$ can be affected by data changes in the covariates (or non-conformity scores), labels, and group memberships.

### F.1    EXAMPLE: DIFFERENTIAL PRIVACY BOUNDS FOR ENHANCED PRIVACY PROTOCOL

Observe using the *enhanced privacy* approach, $\Delta h \leq \frac{1}{n_k}\left(\frac{1}{L^{(g_a,\tilde{y})}} + \frac{1}{U^{(g_b,\tilde{y})}}\right)$. For the *communication efficient* approach $\Delta h \leq \frac{1}{n_k L^{(g_a,\tilde{y})}}$ for the upper coverage term and $\Delta h \leq \frac{1}{n_k U^{(g_b,\tilde{y})}}$ for the lower coverage term. The server will know the sensitivity used by each client and their choice of $\epsilon$ and $\delta$.

To demonstrate how the server can estimate the coverage gap, we will consider an example using the *enhanced privacy* approach. The result from server aggregation is,

$$\text{cov\_gap\_est}(\lambda, F_m, g_a, g_b, \tilde{y})$$
$$= \sum_{k=1}^{K} \gamma_k \underbrace{\left\{\frac{(\alpha_k^{(g_a,\tilde{y});\lambda} + 1)}{(n_k + 1)L^{(g_a,\tilde{y})}} - \frac{\alpha_k^{(g_b,\tilde{y});\lambda} n_k^{(g_b,\tilde{y})}}{(n_k^{(g_b,\tilde{y})} + 1)(n_k + 1)U^{(g_b,\tilde{y})}} + X_k^{(g_a,g_b,\tilde{y})}\right\}}_{\text{Returned by the Client}}, \tag{22}$$

where $X_k^{(g_a,g_b,\tilde{y})} \sim \mathcal{N}(0, \sigma^2_{k;(g_a,g_b,\tilde{y})})$ such that $\sigma^2_{k;(g_a,g_b,\tilde{y})}$ provides $(\epsilon_k, \delta_k)$-DP for the client. Then observe,

$$\text{cov\_gap\_est}(\lambda, F_m, g_a, g_b, \tilde{y})$$
$$= \underbrace{\sum_{k=1}^{K} \gamma_k \left\{\frac{(\alpha_k^{(g_a,\tilde{y});\lambda} + 1)}{(n_k + 1)L^{(g_a,\tilde{y})}} - \frac{\alpha_k^{(g_b,\tilde{y});\lambda} n_k^{(g_b,\tilde{y})}}{(n_k^{(g_b,\tilde{y})} + 1)(n_k + 1)U^{(g_b,\tilde{y})}}\right\}}_{\text{true coverage gap}} + \underbrace{\sum_{k=1}^{K} \gamma_k X_k^{(g_a,g_b,\tilde{y})}}_{\text{Guassian R.V}} \tag{23}$$

$$= \text{cov\_gap}(\lambda, F_m, g_a, g_b, \tilde{y}) + X, \quad X \sim \mathcal{N}\left(0, \sum_{k=1}^{K} \gamma_k^2 \sigma^2_{k;(g_a,g_b,\tilde{y})}\right) \tag{24}$$

Using a prespecified probability $\beta$ we can accept or reject the statement $\text{cov\_gap\_est}(\lambda, F_m, g_a, g_b, \tilde{y}) \leq c$. In other words, we can check whether,

$$\text{cov\_gap}(\lambda, F_m, g_a, g_b, \tilde{y}) + X \leq c \implies X \leq c - \text{cov\_gap}(\lambda, F_m, g_a, g_b, \tilde{y})$$

$$\implies \boxed{\underbrace{\frac{X}{\sqrt{\sum_{k=1}^{K} \gamma_k X_k^{(g_a, g_b, \tilde{y})}}}}_{\text{Standard Normal RV}} \leq \frac{c - \text{cov\_gap}(\lambda, F_m, g_a, g_b, \tilde{y})}{\sqrt{\sum_{k=1}^{K} \gamma_k X_k^{(g_a, g_b, \tilde{y})}}}}.$$

Then, if $\Phi\left(\frac{c - \text{cov\_gap}(\lambda, F_m, g_a, g_b, \tilde{y})}{\sqrt{\sum_{k=1}^{K} \gamma_k X_k^{(g_a, g_b, \tilde{y})}}}\right) > \beta$, where $\Phi$ is the CDF of the standard normal distribution, we can accept the coverage gap as being less than $c$. In other words, with probability $\beta$, the closeness criterion is satisfied with $\lambda$.

While using Gaussian noise results in a PAC-style guarantee, one could instead add strictly positive noise via an exponential mechanism Dwork et al. (2014), where the noise $X \sim \exp(\frac{\epsilon}{2\Delta h})$ is selected to satisfy $\epsilon$-DP, i.e., $(\epsilon, 0)$-DP. This would result in an overestimate of the actual coverage gap. If the overestimate satisfies the closeness criterion, then the server would assert that the exact coverage gap also satisfies the closeness criterion–thus restoring the strict (non-PAC) guarantee in FedCF.

## G  FEDCF FOR AUDITING

Auditing tools are vital for regulatory bodies to ensure ML models comply with fairness and safety standards (Maneriker et al., 2023). In this regard, we present how FedCF can be used to determine if a federated conformal predictor is *fair* according to the regulator's specification of fairness and closeness criterion, $c$ (U.S. Equal Employment Opportunity Commission, 1979; New York City Council, 2021; 2023; European Parliament and Council of the European Union, 2024).

To assess compliance, FedCF can use the global threshold ($\lambda$) values used by the previously trained conformal-predictor and provide it to each client. Then, the client should send the sufficient values calculated via Algorithm 4 (or Algorithm 5) to compute the federated coverage gap. The server would aggregate these values using Algorithm 6. If the calculated coverage gap is below $c$, then the server can assert that the conformal predictor is fair.

Our auditing approach does not require all clients to provide data for auditing. As discussed in Section 3.1, our guarantees hold assuming that the test-point, $(\boldsymbol{x}_{\text{test}}, y_{\text{test}}) \sim \sum_{k=1}^{K} \gamma_k P_k$, is sampled from a mixture of client distributions where $\gamma_k$ is the probability the test point is sampled from $P_k$, or equivilantly is exchangeable with data from client $k$. Thus, if a subset of clients used to train the original federated conformal predictor provides auditing data, then the audit guarantees will hold assuming that $(\boldsymbol{x}_{\text{test}}, y_{\text{test}})$ are sampled from a mixture consisting of the subset of clients used for auditing. This result allows clients to *independently* decide if they would like to submit data for auditing.

The auditing tool provided by FedCF can also be used to ascertain the *marginal* fairness with respect to each client. Using the auditing procedure described above with data from one client, FedCF can determine if the global, federated conformal predictor maintains fairness with respect to data from a single client. If the computed coverage gap is less than $c$, then the fairness guarantees hold with regard to $(\boldsymbol{x}_{\text{test}}, y_{\text{test}}) \sim P_k$, i.e., the client's marginal distribution.

## H    MORE RESULTS

Here, we provide additional results for the ACS and Pokec-{n,z} datasets. Recall, in each figure, we use a **solid** line to represent the *average* efficiency of the **base federated conformal predictors** across different thresholds and a **dashed** line to represent the corresponding *average* worst-case fairness disparity. The bar plot shows the efficiency and worst-case fairness disparity using FedCF, while the **dots** indicate the *desired* fairness disparity. We report the average base performance for clarity and readability

### H.1    IMPACT OF DATA HETEROGENEITY ON ACSEDUCATION: US VS CONTINENTAL US

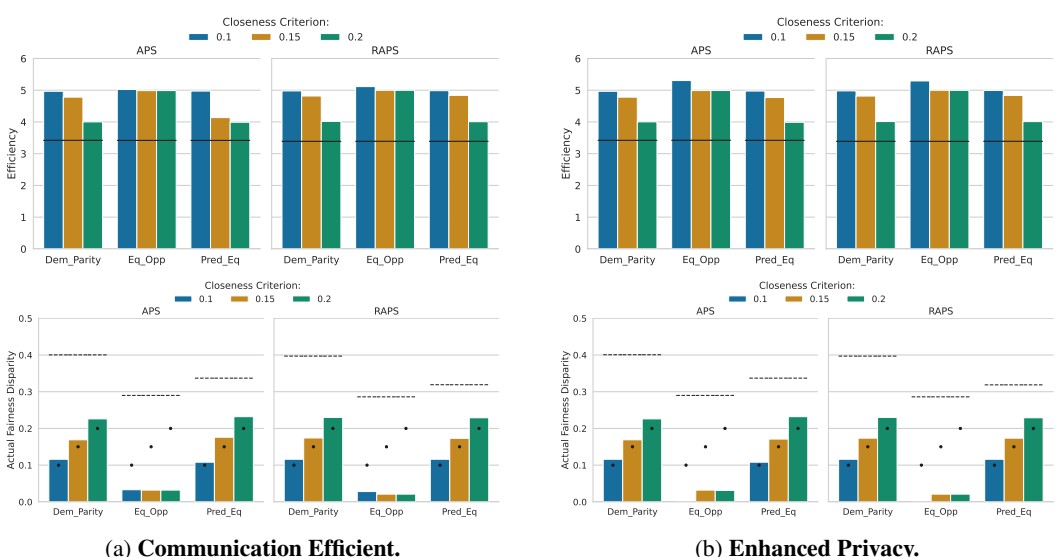

(a) **Communication Efficient.**        (b) **Enhanced Privacy.**

Figure 4: **ACSEducation, Small, Interval Bounds.** The plots in the top row indicate the efficiency with the corresponding fairness disparity plots in the bottom row. We observe that when all US states are included (and Puerto Rico), the closeness criterion is satisfied. However, the efficiency for Equal Opportunity is high for all closeness criterion values, especially compared to the continental US version of ACSEducation in Figure 5. This result stems from a conservative coverage gap estimate during calibration due to limited covariate representation for some groups.

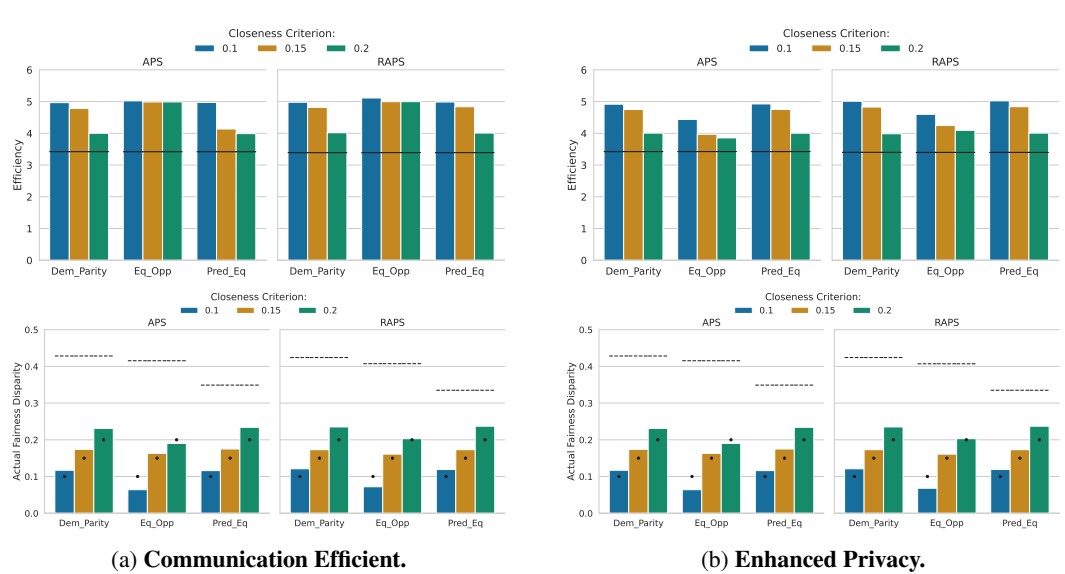

(a) **Communication Efficient.**    (b) **Enhanced Privacy.**

Figure 5: **ACSEducation, Continental Small, Interval Bounds.** The top row demonstrates the efficiency of FedCF when using the continental version of ACSEducation, and its fairness disparity on the bottom row. Compared to Figure 4, the efficiencies improved (particularly for Equal Opportunity using RAPS), due to increased covariate representation for all sensitive groups.

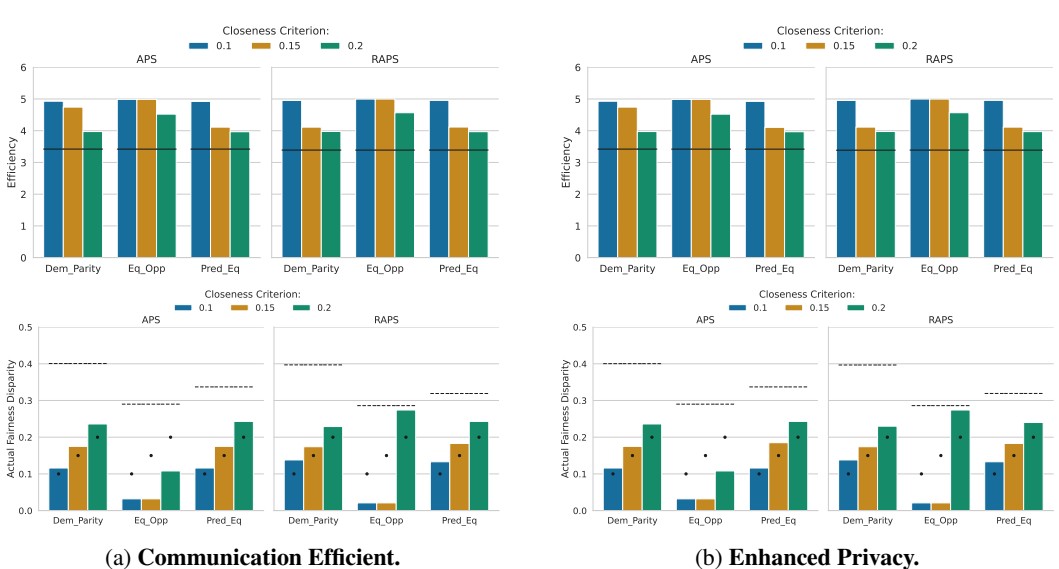

(a) **Communication Efficient.**    (b) **Enhanced Privacy.**

Figure 6: **ACSEducation, Small, Point Estimates** The plots in the top row indicate the efficiency with the corresponding fairness disparity plots in the bottom row. We observe that using point estimates will result in a similar or lower efficiency than using the interval bounds approach in Figure 4, at the cost of a similar or higher fairness violation. Because the MLE does not provide a finite sample guarantee, the violation can exceed the desired closeness criterion, but will be lower than the baseline federated conformal predictor.

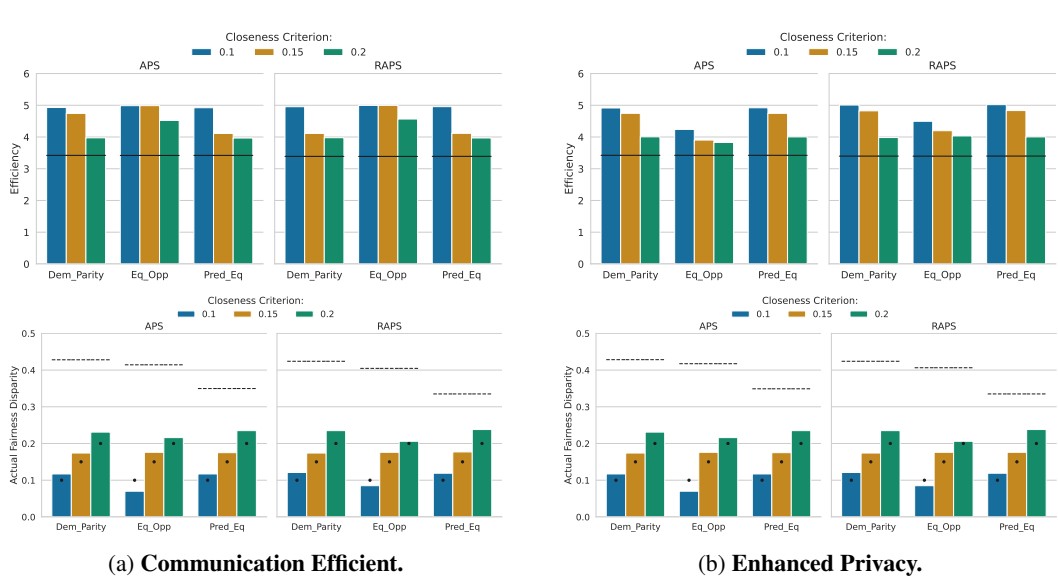

(a) **Communication Efficient.**

(b) **Enhanced Privacy.**

Figure 7: **ACSEducation, Continental Small, Point Estimates.** The plots in the top row indicate the efficiency with the corresponding fairness disparity plots in the bottom row. We observe that using point estimates will result in a similar or lower efficiency than using the interval bounds approach in Figure 5, at the cost of a similar or higher fairness violation. Because the MLE does not provide a finite sample guarantee, the violation can exceed the desired closeness criterion, but will be lower than the baseline federated conformal predictor.

## H.2 IMPACT OF DIFFERENT SENSITIVE ATTRIBUTES FOR POKEC-{N,Z}

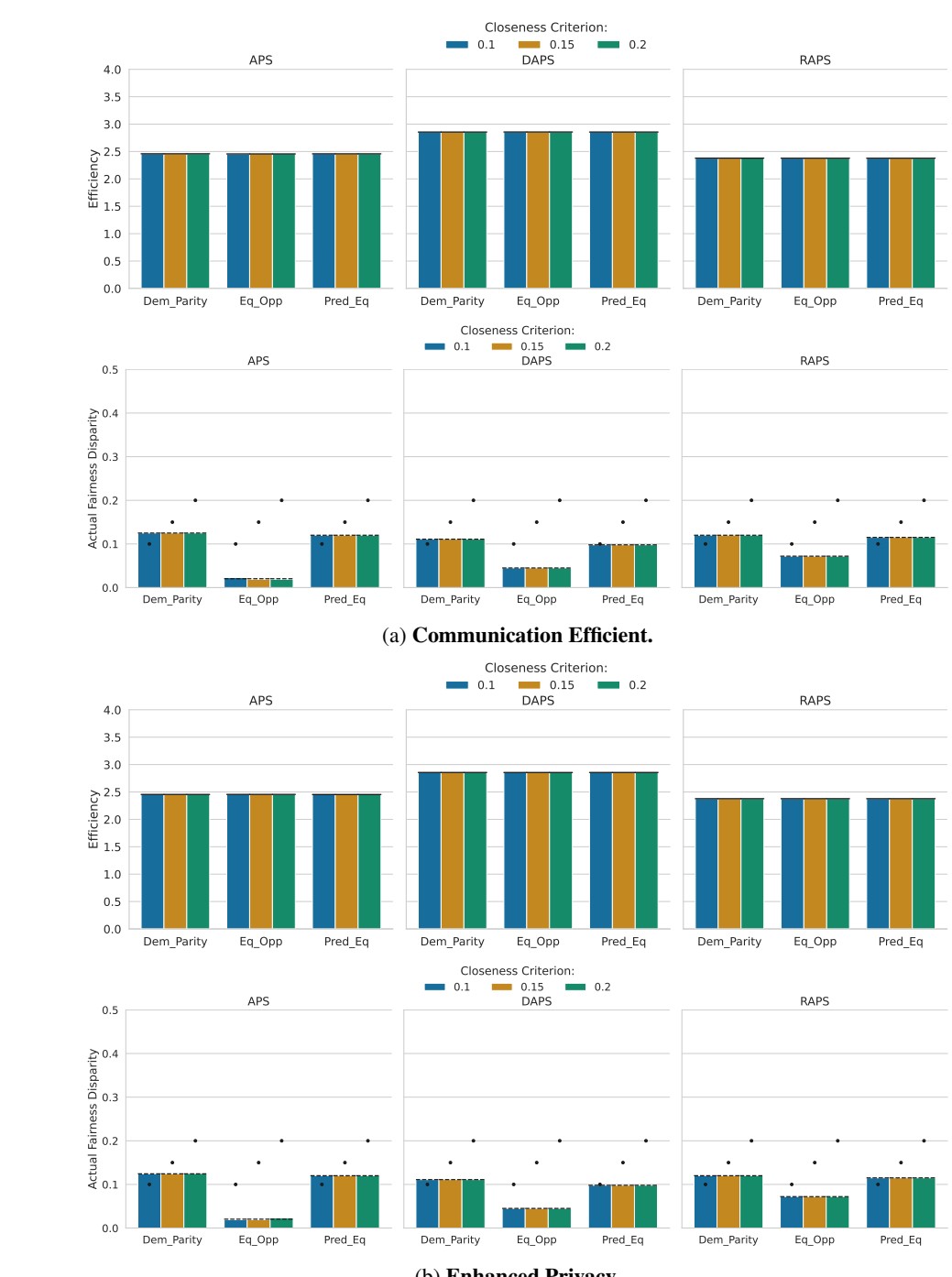

(a) **Communication Efficient.**

(b) **Enhanced Privacy.**

Figure 8: **Pokec-{n,z}**, *gender*. For each plot (a) and (b), the top plots are for the efficiency, and the bottom plots are for the fairness disparity. The baseline disparity is within the closeness criterion, so we see no changes in efficiency when using FedCF. This is the case when using either the *communication efficient* and *enhanced privacy* protocols.

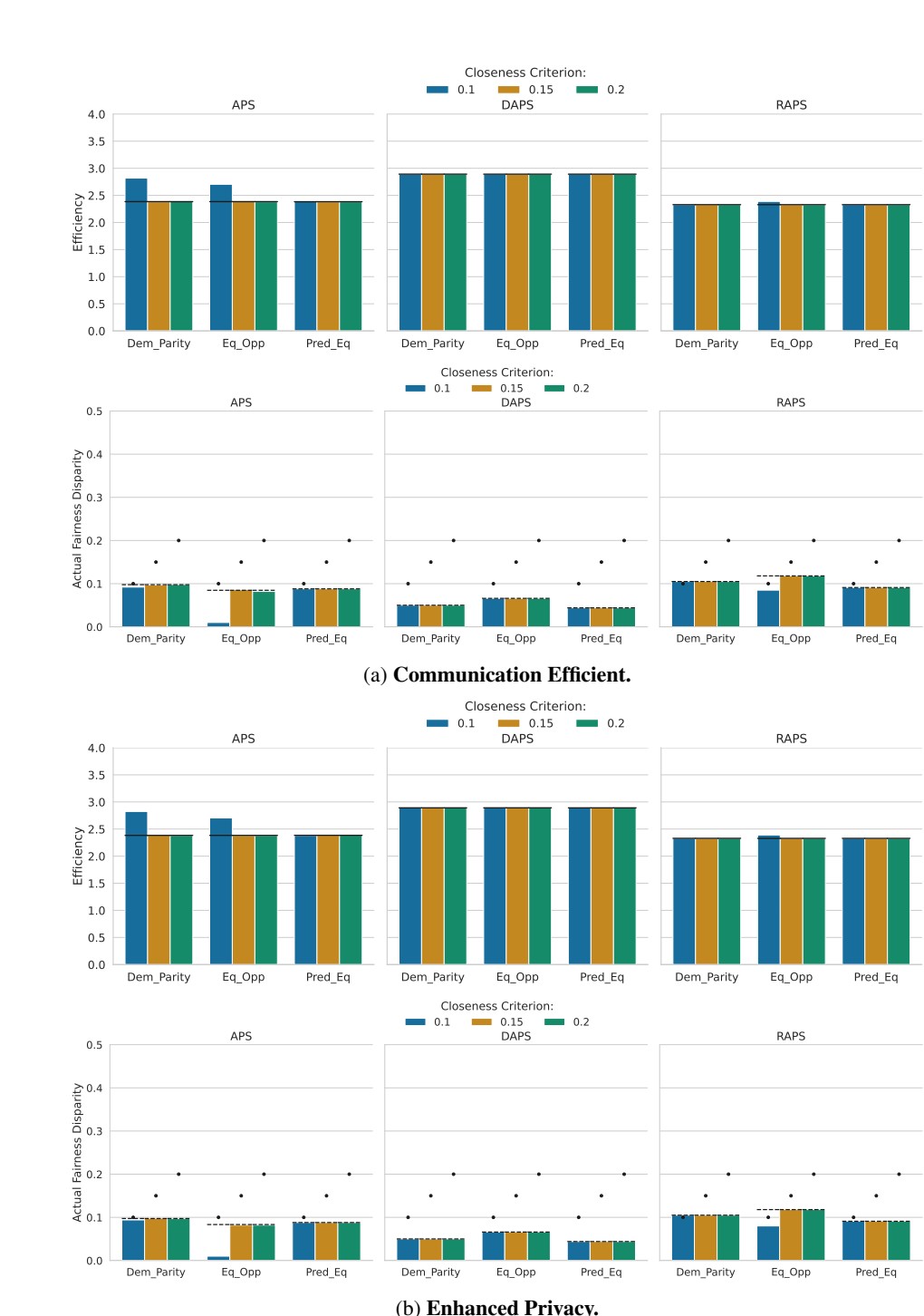

(a) **Communication Efficient.**

(b) **Enhanced Privacy.**

Figure 9: **Pokec-{n,z}**, *region*. For each plot (a) and (b), the top plots are for the efficiency, and the bottom plots are for the fairness disparity. Note that while the baseline disparity is within the closeness criterion for the test set, the finite-sample guarantee from using the interval bounds ensures FedCF looks for a better threshold, resulting in a smaller violation with a small cost to efficiency. This is the case when using either the *communication efficient* and *enhanced privacy* protocols.

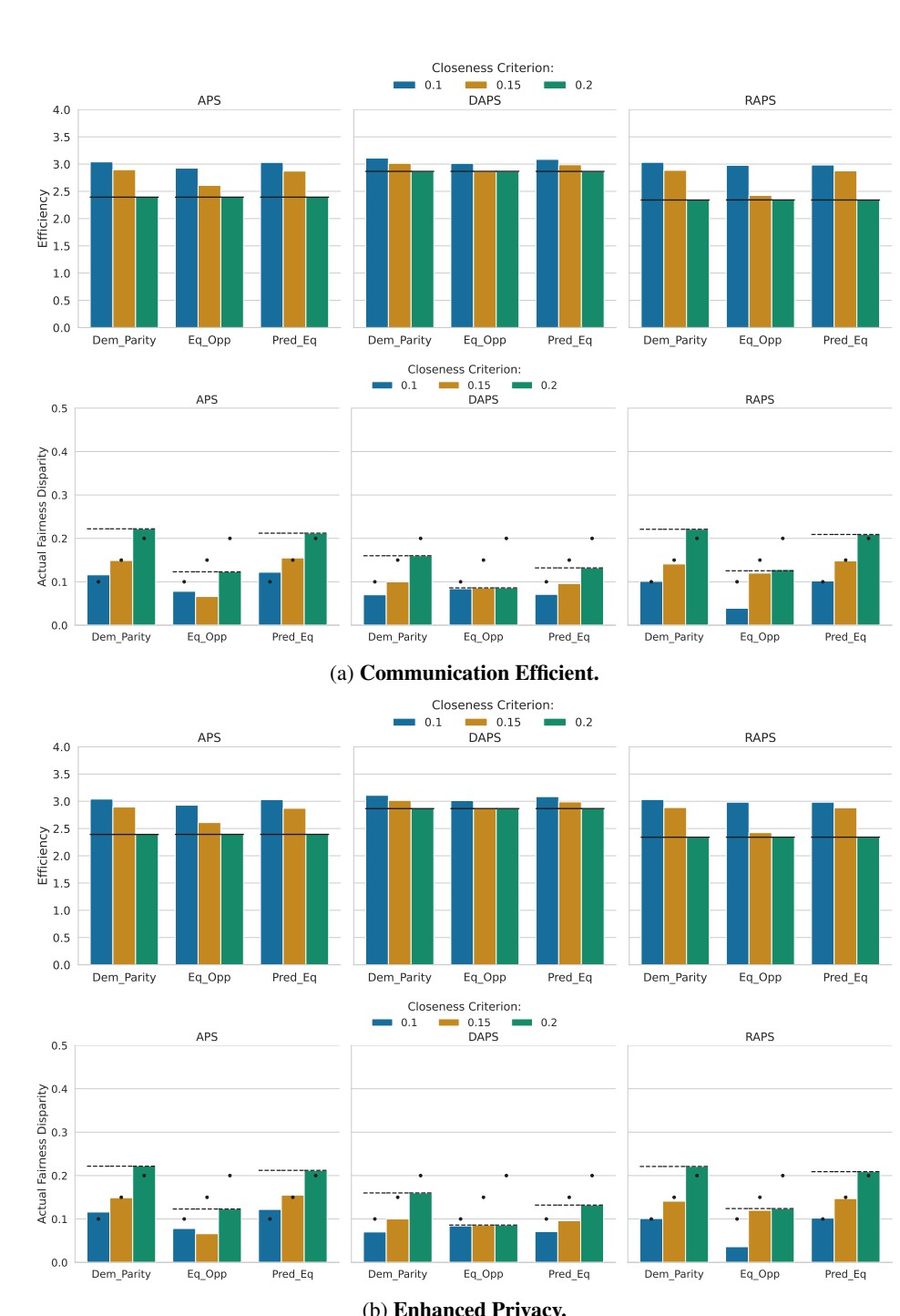

(a) **Communication Efficient.**

(b) **Enhanced Privacy.**

Figure 10: **Pokec-{n,z},** *region* **and** *gender***.** For each plot (a) and (b), the top plots are for the efficiency, and the bottom plots are for the fairness disparity. In the case of intersectional fairness, since there are more groups, the violation will be worse than considering a single sensitive attribute. We observe that in all cases, FedCF produces a threshold that satisfies the closeness criterion, at a slight cost to efficiency. This is the case when using either the *communication efficient* and *enhanced privacy* protocols.

# I ADDITIONAL RESULTS FROM REBUTTAL PHASE

## I.1 JUSTIFICATION FOR DESCENT-BASED APPROACH TO CF

In this experiment, we define the discrete search space needed in (Vadlamani et al., 2025) as $\Lambda = \texttt{linspace}(\hat{q}(\alpha), 2, \texttt{num\_rounds})$, where $\hat{q}(\alpha)$ is the minimal lambda to satisfy $1 - \alpha$ coverage of standard federated CP, and 2 represents the upper bound of the RAPS score (Angelopoulos et al., 2022). We observe that more communication rounds are required for the iterative approach to achieve performance comparable to our descent-based approach. This is because linspace determines the precision of the $\lambda$ values, so by using fewer rounds, we will find a less precise, and in turn more conservative, $\lambda$. This is not a limitation for our descent-based approach, which will continue to converge to a constraint-satisfying $\lambda$. The results in Table 1 below demonstrate that it took 1000 rounds of the Iterative approach to achieve sufficient granularity and match the performance of the Descent-Based approach, which only used 100 rounds. Thus, we demonstrate approximately a $10\times$ speedup to achieve comparable performance, justifying the descent-based approach as a necessary adaptation for the federated learning setting.

Table 8: **ACSIncome (small) using RAPS.** Each entry is of the form, **efficiency/fairness disparity**. We compare the Descent-Based (with 100 communication rounds) and the Iterative method (with 100 and 1000 communication rounds) across different closeness criteria ($c$).

| Method | $c = 0.1$ | $c = 0.15$ | $c = 0.2$ |
|---|---|---|---|
| | Eff / Disp | Eff / Disp | Eff / Disp |
| **Descent-Based** (rounds=100) | 3.127 / 0.096 | 2.977 / 0.179 | 2.816 / 0.257 |
| **Iterative** (rounds=100) | 3.239 / 0.119 | 3.137 / 0.139 | 2.890 / 0.213 |
| **Iterative** (rounds=1000) | 3.120 / 0.137 | 2.989 / 0.170 | 2.821 / 0.251 |

