# OpenReview forum: "FedCF: Fair Federated Conformal Prediction"
_ICLR.cc/2026/Conference — Submitted to ICLR 2026_

### Official Review · Reviewer_EQi5 · 2025-10-30

**Soundness:** 3
**Presentation:** 2
**Contribution:** 2
**Rating:** 2
**Confidence:** 4

**Summary:**

This paper extends the Conformal Fairness (CF) framework [1] to federated learning (FL).  The paper suggests the framework provides a way to “audit” the fairness of FL models.  The extension relies on bounding the coverage of the conformal prediction set as a sum of statistics generated by the clients.  The server does a gradient search for the parameter, lambda, (size of the prediction set) that minimizes the squared error between the model’s coverage gap and a desired closeness level (c).  This threshold is then used by all clients in making their predictions.  The FedCF extension is evaluated on 4 datasets using APS, DAPS, and RPAS conformity measures.  Experimental results verify that FedCF can tradeoff fairness disparity against prediction set size in a manner that is plausible for use in auditing the fairness of FL models.

**Strengths:**

+ The proposed extension of conformal fairness to the federated setting is interesting and timely.

+ Use of gradient search for coverage parameter, lambda, is an improvement over what was done in the original CF paper [1].

**Weaknesses:**

-	The bound on the coverage gap in Thm 3.1 is essential for the paper’s federated extension, but there are no results (experimental or theoretical) estimating how tight the bound is.  One would always expect the bounds obtained in a distributed setting to be worse than those that could be obtained using all the available information.  It is important to know how much one loses in estimating the coverage gap in this manner.

-	The paper is missing any meaningful discussion of model performance.  Based on my reading of the paper, FedCF is a post-processing approach to achieving fairness in FL in which all clients send statistics to the server from which the coverage threshold, lambda, is selected and that threshold is then used by all clients.  Essentially, we achieve fairness by assigning positive outcomes to less qualified individuals, until the target for fairness disparity has been achieved.  This comes at a cost of reducing the model’s performance (accuracy, f-score, whatever) on the entire population.  So the paper’s results and the “audit” are of limited value if one can’t assess how much performance is lost in achieving the fairness objective.   Moreover, it should be possible to estimate what the “optimal” performance would be for a given lambda.  It would be good to see how FedCF performance fairs with respect to this optimal cost.

**Questions:**

Questions:

1)	How tight is the bound in Thm 3.1 (experimental or theoretical)?

2)	How does model performance decrease with reducing fairness disparity?  How does this achieved performance compare to what is optimally achievable either in a centralized or federated setting?

3)	While this paper focuses on its extension of the CF framework in [1], I was also curious about how it differed from the Federated Conformal Prediction framework in [2]?

Additional Remarks:

1)	I thought the writing was a bit disorganized, since I felt I had to hunt through the paper to understand how FedCF enforces fairness across the clients.  The paper might benefit from some reorganization in the introduction and background sections in a way that builds up more systematically to the FedCF algorithm.

2)	The paper needs to address the model performance lost when the clients use the prediction set threshold determined by the server.

3)	The predict set threshold is used by all clients.  I think it might be useful to examine whether the set thresholds should be client specific.  I suspect this can be one way of recovering model performance lost by having all clients use the same threshold.

References:

[1] Vadlamani, et al. “A generic framework for conformal fairness”, ICLR 2025

[2] Lu, et al., ”Federated conformal predictors for distributed uncertainty quantification”, ICML 2023

---

> ### Author Response · Authors · 2025-11-21
> **Response to reviewer**
>
> We thank the reviewer for taking the time to review our paper.
>
> > [W1/Q1] How tight is the bound in Thm 3.1 (experimental or theoretical)?
>
> We note that although the interval in Theorem 3.1 is deliberately conservative to ensure the same finite-sample guarantees provided by conformal prediction and conformal fairness, the paper also includes an MLE estimate (lines 463 to 469), which serves as a practical proxy for tightness and shows that the interval is not overly loose in realistic regimes.
>
> > [W2/Q2/AR2] How does model performance decrease with reducing fairness disparity?...
>
> In conformal prediction, we treat the underlying model as a black-box and perform uncertainty quantification in a  **post-hoc** manner. CP does not modify the underlying model performance (i.e., accuracy and f-score).
> The two main metrics of CP are coverage and efficiency, which refer to the actual test-time probability that the prediction set covers the true label and the average prediction set size, respectively. We also consider the worst-case fairness disparity, defined as the difference in fairness-constrained conditional coverages.
>
> > [Q3] How does FedCF differ from the Federated Conformal Prediction framework?
>
> The Federated Conformal Prediction framework serves as the baseline for all our experiments. Also, as Conformal Fairness is a fair variant of standard conformal prediction, our FedCF framework is a fair variant of federated conformal prediction.
>
> > [AR1] writing was a bit disorganized…
>
> We are continuously improving our manuscript and are happy to take concrete suggestions from the reviewer.
>
> > [AR3] The prediction set threshold is used by all clients. What about client-specific?
>
> As in [1], assuming test points follow a mixture distribution across clients, we use a single $\lambda$ (or a class-wise $\lambda$ vector) to capture client heterogeneity collectively. If additional information is available about the test point's originating client, we can instead use client-specific $\lambda$ values by applying the standard Conformal Fairness framework to that client's data.
>
> ---
> [1] Lu, et al.," Federated conformal predictors for distributed uncertainty quantification", ICML 2023

---

### Official Review · Reviewer_py2j · 2025-10-31

**Soundness:** 2
**Presentation:** 3
**Contribution:** 2
**Rating:** 4
**Confidence:** 5

**Summary:**

This paper explore fair federated learning by conformal prediction, which is an extension of conformal fairness to the federated learning. They audit a federated model for fairness by analyzing the fairness-related gaps for different demographic groups. Numerical experiments is rich.

**Strengths:**

- The authors introduce a descent-based conformal fairness optimization to solve fair federated learning problems.
- The authors provide a criterion to update learning rate in algorithm.
- The writing is good.

**Weaknesses:**

- There may be some mistakes in the core algorithm--descent based CF optimization (see questions).
- Numerical results are not good enough and the efficiency (the average prediction set size is large).
- This paper is an intersection of conformal federated learning and conformal  fairness algorithms?

**Questions:**

- Descent-based CF optimization, descent or ascent? You are from a very small $\lambda_0$? and $b_{t+1}$ should be positive. Then Lines 10 and 14 is a gradient ascent with momentum.
- Line 8 of Algorithm 1, do you need to return $\lambda_{opt}$ here? If not, $\Delta \lambda$ may be negative. Are there some typos in Line 11 of Algorithm 1 to ensure $\Delta \lambda$ not to be negative.
- I did not understand the dots or dotted line in Figure 1.
- The efficiency is bad compared with baseline.

---

> ### Author Response · Authors · 2025-11-21
> **Response to reviewer**
>
> We thank the reviewer for taking the time to review our paper.
>
> > [W1] Descent vs Ascent
>
> Since our framework aims to **minimize**
> $\lambda$ subject to the fairness specifications, descent is the more appropriate term.
>
> > [W2] Line 8 of Algorithm 1, do you need to return $\lambda_{\text{opt}}$ here? Are there some typos in Line 11 of Algorithm 1 to ensure $\Delta\lambda$ is not negative
>
> In Line 8, we set $\lambda_\text{opt}$ to be the current constraint-satisfying $\lambda$; however, we do not return it since there can be a smaller constraint-satisfying $\lambda$ value. We thank the reviewer for catching the typo in line 11 ($\Delta \lambda$), which should always be positive (and we have corrected $\lambda_0 - \lambda_t$ to $\lambda_t - \lambda_0$).
>
> > [W3] I did not understand the dots or dotted line in Figure 1.
>
> As described in Lines 348-354, the dots represent the target fairness disparity or closeness criterion, $c$. The dashed line indicates the average fairness disparity of the baseline federated conformal predictor. The goal is for the fairness disparity to be at or below the dots, which our method achieves, unlike the baseline (where the dashed line typically lies above both our method and the dots).
>
> > [W4] The efficiency is bad compared with the baseline.
>
> As discussed in Lines 422-431, we acknowledge there is a tradeoff between efficiency and fairness, similar to the tradeoff discussed in the ML fairness literature [1]. Although our method may incur a slight efficiency cost, it meets the specified fairness requirements. We leave it to the regulatory bodies and practitioners to decide how much they are willing to incur.
>
> ---
> [1] Zhao and Gordon, "Inherent Tradeoffs in Learning Fair Representations", JMLR 2023

---

### Official Review · Reviewer_jdor · 2025-10-31

**Soundness:** 3
**Presentation:** 3
**Contribution:** 3
**Rating:** 4
**Confidence:** 3

**Summary:**

In this paper, the authors introduce the Federated Conformal Fairness (FedCF) approach, which extends the Conformal Fairness framework to a federated setting. The paper introduces two communication protocols that allow for a balance between communication overhead and privacy.
Authors conduct experiments on data from different domains, namely tabular, graph, and image tasks, and demonstrate that FedCF can control fairness disparities with acceptable trade-offs in set size.

**Strengths:**

1. Overall, the paper is easy to read, and the idea is clear.

2. The evaluation of the models is reasonable, and the choice of experiments is diverse (but see Weaknesses).

**Weaknesses:**

Some of the points below are not necessary weaknesses, but rather questions that I did not understand during review. Can the authors kindly address them?

1. In experiments, only datasets with a relatively small number of clients are used (at most 51 [line 313]). In practice, however, there could be millions of clients. Even if the performance of the proposed framework is reasonable for the moderate number of clients, it could potentially degrade with K. I think an experiment on a dataset with a large number of clients (e.g., Stack Overflow) can resolve this issue.

2. Another practical challenge for federated learning is partial participation, when clients can be unresponsive from one communication round to another. However, the algorithm appears not to consider this situation. Will this situation create any challenges for the framework? Could this lead to bias?

3. As an additional limitation of experiments, datasets were partitioned to create clients rather than collected in a native FL way. Therefore, the evaluation lacks real-world heterogeneity.

4. This is rather a question. Currently, the approach tunes a single global $\ lambda$ for all clients, which may be inefficient given the real-world heterogeneity among clients. Can one incorporate multiple thresholds (per client or per group of clients) into the framework?

5. In practice, new clients can join the distributed system with their new local datasets. How can the FedCF adopt them?

**Questions:**

See Weaknesses

---

> ### Author Response · Authors · 2025-11-21
> **Response to reviewer**
>
> We thank the reviewer for taking the time to review our paper.
>
> > [W1] … only datasets with a relatively small number of clients are used (at most 51 [line 313]).
>
> To the best of our knowledge, publicly available datasets with a large number of clients (e.g., StackOverflow) are not suitable for evaluating group-fairness calibration. These datasets typically assign each user or entity to its own client, thereby removing sensitive attributes relevant to fairness evaluation, often for privacy reasons. As evidence, we refer the reviewer to the [FedScale datasets repository](https://github.com/SymbioticLab/FedScale/tree/master/benchmark/dataset), where all classification datasets have had their sensitive attributes removed. The few datasets in this repository that retain sensitive attributes correspond to tasks that fall outside the scope of our work, and we leave them for future exploration.
>
> That being said, in real-world deployments where fairness guarantees are most critical (e.g., hospitals, financial institutions), clients represent data-holding entities with adequate sample sizes. Such deployments naturally operate at a client scale comparable to what we evaluate in our work.
>
> If the reviewer is aware of any federated learning datasets for multi-class classification that include sensitive attributes (or reasonable proxies) that we may have overlooked, we would be happy to investigate them.
>
> > [W2] … clients can be unresponsive from one communication round to another…Could this lead to bias?
>
> The FedCF framework works in the same setting as Federated Conformal Prediction (FedCP). As inherited from FedCP, if a client drops out throughout the entire calibration procedure, the mixture model used for partial-exchangeability would not account for this client. If a client drops out during a single communication round, our framework can ignore this round, as we initialize it with a sub-optimal constraint-satisfying solution. Thus, intermittent dropout does not invalidate guarantees (and does not induce biases), but only reduces efficiency if many clients consistently fail to participate.
>
> > [W3]  … evaluation lacks real-world heterogeneity.
>
> For ACSIncome and ACSEducation, which are census-based datasets, we create clients by state or US region, maintaining the natural data heterogeneity. For Pokec, we treat Pokec-n and Pokec-z as two separate clients, as they represent subgraphs of the original, larger Pokec social network, split based on two regions of Slovakia.
>
> > [W4] Can one incorporate multiple thresholds (per client or per group of clients) into the framework?
>
> As in [1], assuming test points follow a mixture distribution across clients, we use a single $\lambda$ (or a class-wise $\lambda$ vector) to capture client heterogeneity collectively. If additional information is available about the test point's originating client, we can instead use client-specific $\lambda$ values by applying the standard Conformal Fairness framework to that client's data.
>
> > [W5] How can FedCF adopt new clients with their new local datasets
>
> FedCF is a post hoc process to FedCP [1], meaning the underlying federated conformal predictor would require recalibration to account for the new client's distribution, and FedCF would need to be reoptimized accordingly. Compared to training the federated model, recalibrating with FedCP and FedCF on the held-out calibration has insignificant computation costs and is thus straightforward.
>
> ---
> [1] Lu, et al.,"Federated conformal predictors for distributed uncertainty quantification", ICML 2023

---

### Official Review · Reviewer_LJua · 2025-11-01

**Soundness:** 2
**Presentation:** 3
**Contribution:** 2
**Rating:** 4
**Confidence:** 4

**Summary:**

The paper introduces FedCF, a framework that brings conformal fairness into federated learning by using coverage decomposition along with two communication protocols. Its main contribution is in adapting and integrating existing conformal-fairness methods for use in federated settings, while offering relatively limited novel algorithmic advances.

**Strengths:**

1. Fair federated learning is an important and timely research direction given the increasing deployment of ML systems in sensitive domains.

2. The experimental results preliminarily demonstrate that the proposed method can control fairness disparities within specified bounds.

**Weaknesses:**

1. The work primarily combines existing methods without substantial algorithmic innovation. While the extension of conformal fairness to federated settings is reasonable, the approach is relatively straightforward and lacks deeper theoretical understanding.

2. The experimental setup is quite far from a realistic federated setting. Since the datasets are split artificially rather than coming from natural sources, the evaluation may fail to reflect the true data heterogeneity that is central to federated learning.

3. The exchangeability assumption across clients creates tension with federated learning principles. Since federated learning is specifically designed to handle heterogeneous, non-IID data distributions, this assumption potentially undermines the method's applicability in practice.

4. Communication costs scale unfavorably with the number of sensitive attributes, particularly for the privacy-preserving variant. This scalability concern may limit practical deployment in settings with multiple fairness considerations.

5. The evaluation lacks comparison with alternative federated fairness approaches, making it difficult to assess relative performance. Additionally, the efficiency degradation observed with fairness constraints, especially using interval bounds, triggers concerns about practical utility.

**Questions:**

1. How does the method handle data heterogeneity in realistic federated environments where the exchangeability assumption may not hold?

2. Can you provide a detailed analysis of communication costs and scalability when the number of clients and sensitive attributes increases significantly?

3. How does FedCF compare against other existing federated fairness approaches in terms of both fairness guarantees and computational efficiency?

---

> ### Author Response · Authors · 2025-11-21
> **Response to reviewer [1/2]**
>
> We thank the reviewer for taking the time to review our paper.
>
> > [W1] The work primarily combines existing methods without substantial algorithmic innovation.
>
> While our work is principled in the ideas of Conformal Fairness (CF) [1], we do provide below several differences from CF to facilitate practical performance in the Federated Learning (FL) setting:
>
> - **Descent-Based CF Formulation:** The original CF algorithm is formulated as an iterative procedure, which becomes inefficient in an FL setting because it requires client–server communication for **every** value of $\lambda$ in the larger $\Lambda$ threshold space (as defined in [1]) until it finds the minimal constraint-satisfying $\lambda$ (which could be very close to the $\sup \Lambda$. In this work, we reformulate CF as a gradient-based descent method, allowing it to be embedded directly into an FL-style algorithm and to integrate naturally with existing FL systems. We will include empirical results to justify our choice of an optimization approach rather than an iterative one.
> - **Enabling CF for Decentralized Operation:** To provide theoretical guarantees in the FL setting, we had to non-trivially restructure how to calculate the coverage gap (from [1]) at the client level and how the server should aggregate client coverage gaps. Formally, our aggregation approach, as shown in Theorem 3.1, used the law of total probability to weight the coverage information provided by each client.
> - **Flexibility within FedCF:** Accounting for FL requirements, including communication overhead, privacy, and performance, we also proposed and evaluated the communication efficiency, enhanced privacy, and MLE approaches. We demonstrated how clients can choose between communication-efficient and enhanced-privacy approaches to tailor performance as seen in Appendix D (Tables 6 and 7). Lastly, we showed that the MLE approach satisfies fairness constraints with minimal effect on the prediction set size, as seen in Figure 2.
>
> > [W2/Q1]  … evaluation may fail to reflect the true data heterogeneity that is central to federated learning.
>
> Our work explicitly details datasets with induced data heterogeneity. Notably, ACSIncome, ACSEducation, and Pokec are not randomly partitioned; we refer the reader to lines 312-317 for details on how we partition the dataset, as well as Appendix C for a more thorough discussion on our experimental setup. Furthermore, we discuss our framework's ability to handle data heterogeneity in lines 450-455 in the paragraph titled "On Data Heterogeneity".
>
> > [W3/Q1] The exchangeability assumption across clients creates tension with federated learning principles.
>
> It is worth noting that pure IID/Exchangeability assumptions may not hold due to data heterogeneity. To get around this, our framework (and theory) assumes **partial-exchangeability**: the notion that the test point $s(x_\text{test}, y_\text{test})$ is exchangeable with the data (and underlying distribution) of a particular client with probability $\gamma_k$. Informally, we only require exchangeability for data within a client (locally), but **not** globally across clients. For a more detailed discussion on partial-exchangeability, we refer the reader to lines 114-119 in our work and Appendix 1 of [2].
>
> > [W4/Q2] Communication costs scale unfavorably with the number of sensitive attributes....
>
> We agree that communication cost is an essential consideration in federated systems. In our setting, the number of sensitive attributes is typically small in practice, and the communication required by FedCF is orders of magnitude smaller than the cost of federated training via FedAvg. As quantified in the submission, the coverage bounds transmitted by FedCF are lightweight and exchanged only during the calibration phase, not throughout model training. Thus, while the communication cost scales with the number of fairness constraints, it remains negligible compared to standard FL workflows and should not limit practical deployment.
>
> ---
> [1] Vadlamani, et al. "A generic framework for conformal fairness", ICLR 2025
>
> [2] Lu, et al., "Federated conformal predictors for distributed uncertainty quantification", ICML 2023
>
> [3] Zhao and Gordon, "Inherent Tradeoffs in Learning Fair Representations", JMLR 2023

---

> > ### Author Response · Authors · 2025-11-25
> > **Additional Empirical Results, in response to [W1]**
> >
> > As promised, we include some empirical results comparing our (descent-based) approach to the original CF [1] (the iterative) approach. We will also include these results in the camera-ready version.
> >
> > In this experiment, we define the discrete search space needed in [1] as $\Lambda = linspace(\hat{q}, 2, num\\_rounds)$, where $\hat{q}$ is the minimal lambda to satisfy $1-\alpha$ coverage of standard federated CP, and 2 represents the upper bound of the RAPS score [4]. We observe that more communication rounds are required for the iterative approach to achieve performance comparable to our descent-based approach. This is because linspace determines the precision of the $\lambda$ values, so by using fewer rounds, we will find a less precise, and in turn more conservative, $\lambda$. This is not a limitation for our descent-based approach, which will continue to converge to a constraint-satisfying $\lambda$. The results in Table 1 below demonstrate that it took 1000 rounds of the Iterative approach to achieve sufficient granularity and match the performance of the Descent-Based approach, which only used 100 rounds. Thus, we demonstrate a ~10x speedup to achieve comparable performance, justifying the descent-based approach as a necessary adaptation for the federated learning setting.
> >
> > ---
> >
> > **Table 1:** ACSIncome (small) using RAPS. Each entry is of the form efficiency/fairness disparity, and we compare the results for different closeness criteria ($c$).
> > | $c$  | Descent-Based (num\_rounds = 100) | Iterative (num\_rounds = 100) | Iterative (num\_rounds = 1000) |
> > | ---- | -------------------------- | ---------------------- | ----------------------- |
> > | 0.1  | 3.127/0.096                | 3.239/0.119            | 3.120/0.137             |
> > | 0.15 | 2.977/0.179                | 3.137/0.139            | 2.989/0.170             |
> > | 0.2  | 2.816/0.257                | 2.890/0.213            | 2.821/0.251             |
> >
> >
> > ---
> > [1] Vadlamani, et al. "A generic framework for conformal fairness", ICLR 2025
> >
> > [4] Angelopoulos, A. N., et al. "Uncertainty Sets for Image Classifiers using Conformal Prediction” ICLR 2021

---

> ### Author Response · Authors · 2025-11-21
> **Response to reviewer [2/2]**
>
> > [W5/Q3] How does FedCF compare against other existing federated fairness approaches in terms of both fairness guarantees and computational efficiency?
>
> The closest baseline in this space is Federated Conformal Prediction (FedCP) [2]. We evaluate our approach against this baseline and demonstrate improved fairness. Of course, we would be happy to discuss with the reviewer more constructive suggestions for meaningful baselines that provide hard guarantees at the intersection of Fairness, Federated Learning, and Uncertainty Quantification (specifically, conformal prediction).
>
> Lastly, in Lines 422-431, we acknowledge there is a tradeoff between efficiency and fairness (and it is not possible to achieve both better efficiency and fairness), similar to the utility-fairness tradeoff discussed in the ML fairness literature [3].
>
> ---
> [1] Vadlamani, et al. "A generic framework for conformal fairness", ICLR 2025
>
> [2] Lu, et al., "Federated conformal predictors for distributed uncertainty quantification", ICML 2023
>
> [3] Zhao and Gordon, "Inherent Tradeoffs in Learning Fair Representations", JMLR 2023

---

### Author Response · Authors · 2025-12-01
**Overall Response to AC [1/2]**

We thank all the reviewers for taking the time to review our paper and for providing us with helpful feedback to improve the presentation of our work.

## Overview of Addressed Concerns

We directly address all of the reviewer-mentioned weaknesses and questions in the individual responses. Below, we explicitly call out and clarify broader and recurring concerns.

> [LJua (W1)] Novelty of FedCF

> [py2j (W3)] Work is at the intersection of conformal FL and conformal fairness

> [EQi5 (W1)] Extending CF Theory, Bound Tightness

We discuss below the necessary adaptations and contributions to facilitate Conformal Fairness (CF) [1], in the Federated Learning (FL) setting. We also summarize the existing results.

- **1. Descent-Based CF Formulation:** CF is formulated initially as an iterative (grid-search) procedure, and is inefficient in an FL setting, which requires client–server communication for **every** value of $ \lambda $ in the $\Lambda$ threshold space (as defined in [1]). In this work, we reformulate CF as a gradient-based descent method and include additional results in Table 8 of Appendix I that demonstrate we can achieve an order-of-magnitude (~10x) speedup while maintaining comparable performance to the iterative method.
- **2. Extending CF Theory to FL:** To provide theoretical guarantees in the FL setting, we non-trivially restructured the coverage-gap calculation (from [1]) at the client level. We also introduced server-side aggregation strategies of client coverage gaps to align with the assumed test-time mixture distribution.
- **3. Flexible Aggregation Protocols and Theoretical Bound Tightness:** We account for FL requirements such as communication overhead, privacy, and performance by proposing the communication-efficient (CE), enhanced privacy (EP), and MLE approaches, respectively. We also offer a hybrid of CE and EP, enabling clients to select either individually, and provide results for CE, EP, and Hybrid in Tables 6 & 7 in Appendix D. Lastly, we showed that the MLE approach satisfies fairness constraints with minimal effect on the prediction set size, as shown in Figure 2, thereby allowing tight bounds on the coverage gap.

> [LJua (W2/Q1); jdor (W3)] Evaluation with Real World Data Heterogeneity

We primarily evaluated FedCF on datasets that naturally exhibit real-world data heterogeneity, as discussed in Lines 312-317 and Appendix C. Notably, ACSIncome, ACSEducation, and Pokec are partitioned by state or region, respecting the native client distributions.

> [jdor (W1)] Evaluated FedCF on a relatively small number of clients

To the best of our knowledge, publicly available datasets with a large number of clients (e.g., Stack Overflow with over 300,000+ clients, as suggested by reviewer jdor) are not suitable for evaluating group-fairness calibration, as they purposefully remove sensitive attribute information, often for privacy reasons.

That being said, in real-world deployments where fairness guarantees are most critical (e.g., hospitals, financial institutions), clients represent data-holding entities with adequate sample sizes. Such deployments naturally operate at a client scale comparable to what we evaluate in our work.

> [py2j (W2)] FedCF has worse efficiency than the baseline

As discussed in Lines 422-431, we acknowledge a tradeoff between efficiency and fairness, similar to those in the ML fairness literature. Although our method may incur a slight efficiency cost, it meets the specified fairness requirements. We leave it to the regulatory bodies and practitioners to decide how much they are willing to incur.

> [EQi5 (W2)] Missing discussion of *base model* performance

In conformal prediction, we treat the underlying model as a black-box and perform uncertainty quantification in a **post-hoc** manner. CP does **not** modify the *base model* performance (i.e., accuracy and f-score).

The reviewer followed up on our response, noting that the work would be more interesting if we could use FedCF to improve the underlying model's performance, which we, in fact, do. As discussed in Appendix G, our FedCF framework can also serve as an auditing tool, similar to state-of-the-art ML fairness auditing tools [2, 3]. Practitioners can use the results of these audits to guide model development. Our classwise-lambda approach pinpoints the specific class causing failure, while per-group metrics identify under-covered groups to guide fairness-concerned model improvements.

> [jdor (W4); EQi5 (W3)] Multiple thresholds (per client or per group of clients)?

As in [4], assuming test points follow a mixture distribution, we use a single $\lambda$ (or a class-wise $\lambda$ vector) to capture heterogeneity across clients. If additional information is available about the test point's originating client, we can instead use client-specific $\lambda$ values by applying the standard Conformal Fairness framework to that client's data.

---

### Author Response · Authors · 2025-12-01
**Overall Response to AC [2/2]**

## Summary of Reviewer Interactions
[LJua]: This reviewer had several questions on FedCF, which we addressed in our rebuttal. The questions prompted us to update parts of the manuscript to present additional results and clarify the novelty of our work. While the reviewer did not respond, our responses addressed their concerns and warranted a review of their score.

[jdor] This reviewer gave our work high ratings for *soundness*, *presentation*, and *contribution*, and also explicitly noted that what they included in the “Weaknesses” section were not necessarily weaknesses, but questions or misunderstandings. In light of this, the review was positive, even though their final rating does not reflect this sentiment.

[py2j]: This reviewer’s nominal feedback suggests the core novelties of our work were overlooked and is at odds with their confidence rating of 5. The reviewer raised weaknesses/questions focused predominantly on clarifications, which we addressed in our rebuttal and updated the manuscript accordingly.

[EQi5]: This reviewer seemed to have misunderstood the role conformal prediction (CP) plays in the model lifecycle, in that it is a post-hoc process not directly concerned with the underlying base model performance. After our rebuttal, EQi5 increased their score from $ 2 \to 4 $, and noted that they would have found the work more interesting if we had a way to use our framework to guide model development, which we further addressed.

## Presentation Updates
Based on feedback from all reviewers, we have tightened the writing of our Key Contributions and the presentation of the descent-based formulation of CF, and we have included the new results requested by the reviewers in a separate Appendix I. Specific updates are highlighted in $ \textcolor{blue}{\text{blue}} $ text. Headers are $ \textcolor{blue}{\text{blue}} $ if the entire section is updated. This will be reverted to black for the camera-ready version.

### Roadmap of Appendices
More details about FedCF are available in the appendices.
- A. provides a notation table for clarity.
- B. provides proofs for Theorem 3.1 and the corresponding lemmas.
- C. provides experiment details, including the splitting scheme to preserve data heterogeneity and definitions of non-conformity scores.
- D. talks about the enhanced privacy aggregations strategy, compares results with the communication-efficient aggregation strategy, and provides results using a hybrid of the improved privacy and communication-efficient approach.
- E. provides the complete FedCF algorithm, including all aggregation approaches.
- F. discusses extensions of FedCF to integrate differential privacy.
- G. talks about using FedCF as an auditing tool and how it can be used to guide fair model development, as discussed with Reviewer EQi5.
- H. provides an extended set of results exploring different data heterogeneity schemes.
- I. includes an additional result for FedCF from the rebuttal and will be moved to the main body for the camera-ready version.

## New Results
In response to reviewer LJuA, we provided an empirical comparison of our descent-based CF approach to the iterative approach from [1]. Since our approach does not require a discretized search space, we can find a smaller (more precise) constraint-satisfying $\lambda$ in significantly fewer rounds. The results in Table 8 in Appendix I of our updated manuscript demonstrate a ~10x speedup to achieve comparable performance, justifying the descent-based approach as a necessary adaptation for the federated setting.


---
[1] Vadlamani, et al., “A generic framework for conformal fairness”, ICLR 2025

[2] Maneriker, et al., “Online fairness auditing through iterative refinement”, KDD 2023

[3] Bastani, et al., “Probabilistic verification of fairness properties via concentration”, OOPSLA 2019

[4] Lu, et al., “Federated conformal predictors for distributed uncertainty quantification”, ICML 2023

---

### Meta-Review · Area_Chair_YgCc · 2026-01-04

**Summary:**

1. The proposed FedCF is a post-hoc analysis of the federated model's fairness. This weakens the value relative to analyses used to improve model fairness.
2. The efficiency of FedCF is worse than the baseline

**Reviewer Concerns:**

Reviewer jdor and EQi5's concern about the theory and model has been addressed. The contribution concern and novelty concern are still outstanding.

**Reviewer Scores:**

Reviewer EQi5 would increase its score from 2 to 4

---

### Decision · Program_Chairs · 2026-01-26

Reject